# Structural basis of the dynamic human CEACAM1 monomer-dimer equilibrium

Amit K. Gandhi [1,11✉], Zhen-Yu J. Sun [2,11], Walter M. Kim[1,11], Yu-Hwa Huang[1,11], Yasuyuki Kondo[1,9], Daniel A. Bonsor[3], Eric J. Sundberg[3,4,5,10], Gerhard Wagner [6], Vijay K. Kuchroo[7], Gregory A. Petsko[8] & Richard S. Blumberg [1✉]

Human (h) carcinoembryonic antigen-related cell adhesion molecule 1 (CEACAM1) function depends upon IgV-mediated homodimerization or heterodimerization with host ligands, including hCEACAM5, hTIM-3, PD-1, and a variety of microbial pathogens. However, there is little structural information available on how hCEACAM1 transitions between monomeric and dimeric states which in the latter case is critical for initiating hCEACAM1 activities. We therefore mutated residues within the hCEACAM1 IgV GFCC′ face including V39, I91, N97, and E99 and examined hCEACAM1 IgV monomer-homodimer exchange using differential scanning fluorimetry, multi-angle light scattering, X-ray crystallography and/or nuclear magnetic resonance. From these studies, we describe hCEACAM1 homodimeric, monomeric and transition states at atomic resolution and its conformational behavior in solution through NMR assignment of the wildtype (WT) hCEACAM1 IgV dimer and N97A mutant monomer. These studies reveal the flexibility of the GFCC′ face and its important role in governing the formation of hCEACAM1 dimers and selective heterodimers.

[1] Division of Gastroenterology, Department of Medicine, Brigham and Women's Hospital, Harvard Medical School, Boston, MA, USA. [2] Department of Cancer Biology, Dana-Farber Cancer Institute, Boston, MA, USA. [3] Institute of Human Virology, University of Maryland School of Medicine, University of Maryland, Baltimore, MD, USA. [4] Department of Medicine, University of Maryland School of Medicine, University of Maryland, Baltimore, MD, USA. [5] Department of Microbiology and Immunology, University of Maryland School of Medicine, University of Maryland, Baltimore, MD, USA. [6] Department of Biological Chemistry and Molecular Pharmacology, Harvard Medical School, Boston, MA, USA. [7] Evergrande Center for Immunologic Diseases and Ann Romney Center for Neurologic Diseases, Harvard Medical School and Brigham and Women's Hospital, Boston, MA, USA. [8] Ann Romney Center for Neurologic Diseases, Department of Neurology, Brigham and Women's Hospital, Harvard Medical School, Boston, MA, USA. [9] Present address: Division of Gastroenterology, Department of Internal Medicine, Graduate School of Medicine, Kobe University, Kobe, Japan. [10] Present address: Department of Biochemistry, Emory University School of Medicine, Atlanta, GA, USA. [11] These authors contributed equally: Amit K. Gandhi, Zhen-Yu J. Sun, Walter M. Kim, Yu-Hwa Huang. ✉email: agandhi2@bwh.harvard.edu; rblumberg@bwh.harvard.edu

Carcinoembryonic antigen-related cell adhesion molecule 1 (CEACAM1), a member of the carcinoembryonic antigen cell adhesion molecule (CEACAM) family of glycosylated immunoglobulin (Ig) molecules[1], is expressed on the surface of several cell types where it plays critical roles in morphogenesis[2], apoptosis[3], angiogenesis[4], cell proliferation[5] cell motility[6], fibrosis[7], and most recently as an immunoreceptor important in mediating immune T cell tolerance[8]. Human CEACAM1 (hCEACAM1) is a single pass type I transmembrane protein expressed as 12 alternatively spliced isoforms that all contain an N-terminal V set fold of the immunoglobulin superfamily (IgV) ectodomain followed by up to three type 2 constant immunoglobulin (IgC2) ectodomains (A1, B, A2), a transmembrane sequence, and a signaling cytoplasmic domain (Supplementary Fig. 1). Depending on splice variation, the cytoplasmic domain either includes a long (L) sequence inclusive of two immunoreceptor tyrosine-based inhibitory motifs (ITIMs) or a short (S) domain devoid of ITIMs[9] that impart intracellular inhibitory or non-inhibitory signals, respectively.

CEACAM1 function is triggered by intercellular or *trans* binding of the IgV domain, resulting in higher order surface CEACAM1 oligomerization and subsequent intracellular signal transduction. In contrast to other immunoreceptors such as the T cell inhibitory and mucin domain-containing protein 3 (TIM-3), programmed cell death protein 1 (PD-1), and programmed death-ligand 1 (PD-L1), CEACAM1 serves as its own primary ligand, owing to high affinity homophilic interactions of its unique IgV domain, as well as an important microbial receptor[10]. At basal steady state, CEACAM1 alternates between monomeric and *cis* homodimeric forms on the cell surface[11], thus presenting a conundrum for *trans* interactions due to the requirement of an accessible CEACAM1 monomer and more specifically, an exposed IgV domain for ligand binding. Therefore, CEACAM1 must undergo a dynamic process of *cis* monomer–dimer exchange and *trans* dimer-higher order oligomerization for productive CEACAM1 activation. At present, the structural details of the monomer–dimer-higher order oligomer exchange mechanism are not well understood.

The hCEACAM1 IgV domain contains 108 amino acids arranged in 9 beta strands (ABCC′C″DEFG) that fold into the conserved IgV anti-parallel beta-sandwich tertiary structure adopted by other IgV-containing immunoreceptors including TIM-3, PD-1, and PD-L1[8,12–14]. The opposing ABED and GFCC′ faces of the CEACAM1 beta-sandwich are tethered by an internal salt bridge (R64:D82) that mimics a stabilizing covalent disulfide linkage found in most Ig domains[8]. Although the ABED surface is exclusively glycosylated, CEACAM1 has been suggested to exist in diverse oligomeric states[15] that include an ABED-mediated homodimer[12], but the more dominant oligomeric form appears to be the high-affinity GFCC′-mediated homodimer[8,15] that is conserved among other IgV domain-containing proteins. Unique to the hCEACAM1 IgV GFCC′ surface is the prominent protrusion of the CC′ loop that differs notably from the ordered β-hairpin observed in other described IgV structures. The displaced CC′ loop forms a cleft with the FG loop that exposes the key residues F29, S32, Y34, V39, G41, Q44, Q89, I91, N97, and E99 critical in mediating homophilic CEACAM1 interactions[8,12,13,16]. As demonstrated in our previously reported high resolution (2.04 Å) crystal structure of the wildtype (WT) hCEACAM1 homodimer (PDB code 4QXW,)[8], the side chains of residues S32, Y34, Q44, Q89, N97, and E99 form a hydrogen bonding network at the GFCC′ interface that includes additional side-chain to main-chain backbone interactions between S32 to L95, Q44 to N97, and E99 to G41 and hydrophobic interactions by residues F29, V39, and I91 (Supplementary Fig. 2a–d). However, while these residue-level hydrogen bonded (total 17) and hydrophobic interactions

determine specificity of the homodimerization interface, they have been recently reported to impart varied free energy contributions to the strength of the interaction[16], raising intrigue in the underlying mechanisms of CEACAM1 IgV monomer–dimer exchange.

Although the major mode of CEACAM1 binding is homophilic, several other host and microbial ligands also exist. The binding of cell surface CEACAM1 by these ligands induces higher order multimerization and in the case of microbial ligands, hijacks the downstream signaling machinery to achieve survival gain. Surprisingly, all of the described CEACAM1 host and microbial ligand interactions where they have been defined involve the GFCC′ surface on CEACAM1, thereby requiring disruption of CEACAM1 homophilic interactions to allow for participation in heterophilic interactions. Although the hCEACAM1 IgV domain has high sequence similarity with other hCEACAM family members (Supplementary Tables 1–3), only CEACAM5 appreciably binds to CEACAM1 owing to sequence conservation of its GFCC′ surface, including the CEACAM1-homodimerization dependent residues F29, S32, V39, R43, Q44, I91, and E99[15,17]. More recently, the N-terminal IgV domain of hTIM-3 was demonstrated to bind hCEACAM1 also through GFCC′-mediated interactions[8,13,18]. Similarly, PD-1 has been implicated as a ligand for CEACAM1[19].

Despite the requirement of competing with CEACAM1 as a ligand, several pathogens, including *Escherichia coli*[17], *Neisseria sp.*[20], *Moraxella catarrhalis*[21], *Haemophilus influenza*[22], *Helicobacter pylori*[23], *Fusobacterium sp.*[24], *Candida sp.*[25], and the coronavirus murine hepatitis virus[26], have evolved structurally distinct microbial receptors that universally disrupt CEACAM1 homophilic interactions at the GFCC′ surface to form unique heterodimeric interactions[27]. Recent studies on the *H. pylori* surface protein HopQ demonstrated that HopQ disrupts the hCEACAM1 IgV homodimer by outcompeting homophilic hCEACAM1 IgV GFCC′ interactions ($K_D = 450$ nM) through up to 20-fold higher affinity heterophilic interactions ($K_D = 23–279$ nM) targeting the GFCC′ surface[16,28]. The crystal structure of the hCEACAM1 IgV-HopQ complex demonstrated direct involvement of the hCEACAM1 GFCC′ surface, however, there are considerable structural and biochemical features that distinguish hCEACAM1 IgV-HopQ heterodimerization from hCEACAM1 homodimerization[16]. One specific discriminating feature involves residue N97, which has been reported to nearly abrogate hCEACAM1 homodimerization ($K_D$ ~1 mM) but does not much affect HopQ binding[16]. This observation raises the question about the role of specific residues at the GFCC′ face in determining hCEACAM1 homophilic and heterophilic interactions and highlights the need to decipher the underlying structural and biochemical features that determine hCEACAM1 monomer-homodimer exchange, which enables the formation of various interactions. At present, the role of the GFCC′ face and specific residues in determining the basal monomer–dimer equilibrium at steady state and moreover the conformational behavior of hCEACAM1 in solution still remain elusive.

Here we describe the structural and biochemical features of specific hCEACAM1 mutants present at the GFCC′ surface (V39A, I91A, N97A, E99A) in static conformation by X-ray crystallography and in solution by nuclear magnetic resonance (NMR) spectroscopy. The unique crystal structures reveal a range of subtle and gross conformational changes in the CEACAM1 homodimer interface and highlight the significance of each examined residue. Furthermore, dynamic NMR studies of the N97A mutant substantiate the critical role of this residue in mediating CEACAM1 homodimerization. These studies illuminate the mechanisms that govern dynamic CEACAM1 homodimerization, exploitation of CEACAM1 as a heterophilic ligand and inform therapeutic interventions to target CEACAM1.

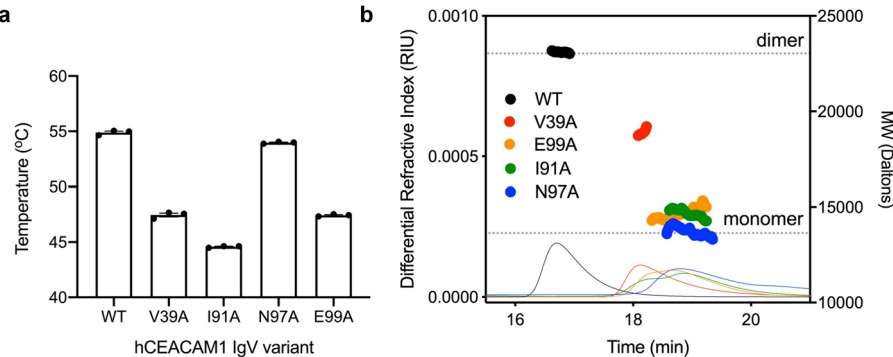

**Fig. 1 Biophysical characterization of CEACAM1 IgV mutants.** Thermal stability and molecular size analysis of hCEACAM1 WT and GFCC′ face mutants. **a** Variations in melting point temperature ($T_M$) determined by differential scanning fluorimetry (DSF) are shown for WT and mutant hCEACAM IgV. **b** Size exclusion chromatography and multi-angle light scattering (SEC-MALS) differential refractive index (dRI) chromatograms and calculated molecular weights are displayed for WT (black), V39A (red), I91A (green), N97A (blue), and E99A (orange).

## Results

**Biophysical characterization of the hCEACAM1 IgV GFCC′ face mutants.** In order to probe the role of the GFCC′ surface, hydrogen bonding network, and hydrophobic interactions in determining the monomer–dimer equilibrium, we introduced alanine substitutions at residues V39, I91, N97, and E99, which have been described to be important for CEACAM1 IgV homo-dimerization and identified as sites for naturally occurring single nucleotide polymorphisms (SNPs) [rs772794650 (I91M), rs1335884800 (N97T), and rs142826356 (E99G)]. We first expressed and purified WT and site-specific mutant hCEACAM1 IgV proteins using our published protocols[13] and measured variations in their respective thermal denaturation temperature ($T_M$), reflective of their stability by differential scanning fluorimetry (DSF) (Fig. 1a). We observed a single $T_M$ for each protein at 25 μM, suggestive of a single step denaturation event (Supplemental Fig. 3) despite whether the protein was expected to be a hCEACAM1 IgV monomer (N97A) or dimer (WT). There was also a direct correlation of $T_M$ with dimerization affinity of the different hCEACAM1 mutants[16,17] suggesting that hCEACAM1 IgV homodimerization stabilizes the IgV domain. One exception was the N97A variant that has been reported to be monomeric[16] but exhibited a similar melting temperature (54.09 °C) compared to WT protein (55.09 °C), suggesting a unique stabilizing property of an alanine at that position and/or promotion of a monomeric state. Next, we assayed the solution characteristics of each hCEACAM1 IgV sequence variant by analytical size exclusion chromatography and multi-angle light scattering (SEC-MALS) and calculation of absolute molecular weight. Each hCEACAM1 IgV variant (100 μM) eluted as a single dominant calculatable molecular weight species but with varying molecular weights ranging from dimer (WT, 23.1 kDa) to monomer (N97A, 13.5 kDa) (Fig. 1b). The presence of a single discernable species for each protein variant and varying intermediate absolute molecular weights suggests rapid rates of exchange between monomeric and dimeric states of the IgV domain rather than a slow equilibrium within the experimental time scale.

**Crystal structures of hCEACAM1 IgV mutants.** To determine the impact of V39, I91, N97, and E99 on hCEACAM1 homodimer formation, we solved the crystal structures of individual V39A, I91A, N97A, and E99A mutant IgV domains to 1.9, 3.1, 1.8, and 1.9 Å resolution, respectively (Table 1), and quantified interactions at GFCC′ and ABED faces (Supplementary Tables 4–12).

The E99A mutant structure revealed a GFCC′ face-mediated homodimer structure (Fig. 2a–d) globally similar to the WT homodimer but with localized conformational differences resulting in a C-alpha root mean square deviation (RMSD) of 0.3 Å (over 1539 atoms) (Fig. 2a, Supplementary Fig. 4a, c). Interestingly, fewer hydrogen bonds (12 vs 17 for E99A and WT, respectively) and weaker hydrophobic interactions were observed for E99A homodimer (Fig. 2a–d, Supplementary Figs. 2a–d, 4a, c). Specifically, side chain to main-chain backbone interactions between E99-G41 were abrogated (Fig. 2a–b, d, Supplementary Fig. 4c) and the intermolecular hydrogen bond network between residues Q89-Y34, N97-Y34, and Q89-N97 (using nomenclature convention here and after, where Q89 residue is from molecule (a) and Y34 residue in *italics* is from molecule (b) present in the crystal asymmetric unit) were disrupted at the E99A mutant homodimer interface (Fig. 2d). In addition, the distance between two opposing hydrophobic valine (V39) residues was slightly higher in the E99A homodimer (3.9 Å) compared to WT (3.7 Å) (Fig. 2b–c).

The low resolution (3.1 Å) of the I91A mutant structure limits atomic level comparison with the WT homodimer and therefore provides a more global assessment on the structural properties of the I91 residue. The I91A IgV domain adopts a GFCC′-mediated homodimer with RMSD of 0.6 Å (over 1489 atoms) compared to the WT homodimer (Fig. 3a, Supplementary Fig. 4b, d) with fewer hydrogen bonded (12 vs 17 for I99A and WT, respectively) and weaker hydrophobic interactions (specially for residue F29) (Fig. 3a–d, Supplementary Figs. 2a–d, 4b, d). Thus, the loss of important hydrogen bond interactions and possibly weaker hydrophobic interactions observed in the E99A and I91A mutant structure support the weak dimeric nature of these mutants as observed in our biophysical studies and previous reports[17,29].

The V39A mutant crystal contained two copies of a hCEACAM1 GFCC′ face–mediated homodimer in the asymmetric unit (Fig. 4a). The first hCEACAM1 V39A dimer (Supplementary Fig. 5a, b) comprising molecules (a) and (b) resembled a WT homodimer with RMSD of 0.7 Å (over 1482 atoms), whereas the second GFCC′-mediated dimer comprising molecules (c) and (d) featured important conformational differences across several beta strands and loops (Fig. 4b) with RMSD of 3.0 Å (over 1655 atoms) compared to the WT homodimer. The major distinguishing feature of this second V39A homodimer structure compared to that of the WT was the increased separation between interacting CC′ loops (10.0 Å vs 3.7 Å) (Fig. 4a, b). Further, we observed a considerable decrease in hydrogen bond (5) and weaker hydrophobic interactions at the GFCC′ face in the V39A homodimer (Figs. 4b, c, 5a, b), consistent with a less stable and less interactive GFCC′ face that possibly reflects an hCEACAM1 IgV monomer–dimer exchange transition state.

**Table 1 Data collection and refinement statistics (molecular replacement).**

| | V39A (PDB code 6XNW) | I91A (PDB code 6XNT) | E99A (PDB code 6XNO) | N97A (PDB code 6XO1) |
|---|---|---|---|---|
| *Data collection* | | | | |
| Space group | P 3 | P 4212 | P 4212 | C 2221 |
| Cell dimensions | 91.4, 91.4, 64.4, 90.0, | 102.1, 102.1, 61.0, 90.0, | 106.8, 106.8, 62.2, 90.0, | 55.9,56.8,124.5, 90.0, |
| a, b, c (Å) | 90.0, 120.0 | 90.0, 90.00 | 90.0, 90.00 | 90.0, 120.0 |
| α, β, γ (°) | | | | |
| Resolution (Å) | 39.59–1.90 (1.94–1.90) | 72.21–3.1 (3.31–3.1) | 75.5–1.9(1.94–1.90 | 28.76–1.76 (1.8–1.76) |
| $R_{merge}$ (%)' | 18.8 (155.1) | 24.6(75.4) | 11.9 (140.3) | 8.6 (833) |
| I/ σI | 7.8 (1.4) | 9.4 (3.8) | 14.3 (2.3) | 19.5 (2.0) |
| Completeness (%) | 100 (100) | 100 (100) | 100 (100) | 96.1(73.2) |
| Redundancy | 7.7 (7.6) | 11.7 (11.7) | 13.6 (13.3) | 11.9 (6.2) |
| *Refinement* | | | | |
| Resolution (Å) | 39.59–1.90 (1.94–1.90) | 72.21–3.1 (3.31–3.1) | 75.5–1.9(1.94–1.90 | 28.76–1.76 (1.8–1.76) |
| No. reflections[a] | 366913 (23237) | 73320 (12990) | 393727(24522) | 231087 (6712) |
| $R_{work}$/$R_{free}$ | 14.5 / 18.6 | 22.1 / 25.8 | 18.9 / 22.3 | 19.2/24.0 |
| *No. atoms* | | | | |
| Protein | 3356 | 1676 | 1674 | 1676 |
| Ligand/ion | 2 | 40 | 54 | 14 |
| Water | 15 | 4 | 86 | 181 |
| *B-factors* | | | | |
| Protein | 28.10 | 42.48 | 30.26 | 15.15 |
| Ligand/ion | 21.17 | 63.02 | 55.84 | 34.20 |
| Water | 26.28 | 33.46 | 38.15 | 23.60 |
| *R.m.s.deviations* | | | | |
| Bond-lengths (Å) | 0.021 | 0.014 | 0.021 | 0.019 |
| Bond-angles (°) | 2.27 | 1..85 | 2.15 | 1.91 |

[a]Values in the parentheses are for highest resolution shell.

Interestingly for both of the V39A-containing dimers, a considerable metal ion electron density (4.0 σ intensity in the 2Fo–Fc electron density map) was observed at V39A molecule (a) and (c); the density was modeled as nickel ($Ni^{++}$) that was hexa-coordinated by residues H105 and V106 from three neighboring IgV molecules found in the unit cell and crystallographic symmetry mates (Supplementary Fig. 6). Similar $Ni^{++}$ binding was observed in a previously published hCEACAM1 WT structure (PDB code 2GK2) where minor interactions between two hCEACAM1 molecules through the ABED face were observed. To our surprise, we observed a similar ABED face-mediated interaction with conserved hydrogen bond interactions between molecules (b) and (c) in the V39A crystal compared to the WT structure (PDB code 2GK2) with comparable similarity (RMSD of 2.7 Å over 1647 atoms) (Supplementary Fig. 7, Supplementary Tables 9, 12). Taken together, the V39A crystal structure demonstrated a notably diminished set of intermolecular interactions at the GFCC′ face but not at the ABED face and provides a structural basis for the monomer-inducing feature of the V39A substitution suggested previously[17,29].

Another defining feature of the GFCC′ interface in the hCEACAM1 WT homodimer is the important contribution of N97 in mediating a critical hydrogen bond network comprising seven hydrogen bond interactions (Supplementary Fig. 2b). It is important to note that we observed that interactions mediated by N97 on molecules (a) and (b) are not perfectly symmetrical. Whereas residue N97 of molecule (a) was observed to interact with S32, Y34, and Q44 residues of molecule (b), N97 of molecule (b) interacted with residues S32, Y34, Q44 but additionally residue Q89 of molecule (a) (Supplementary Fig. 8a). In the N97A mutant crystal, we observed two monomeric N97A molecules [(a), (b)] in the asymmetric unit devoid of any GFCC′-mediated interactions despite high concentration (>800 μM) and global similarity in secondary structure of both N97A molecules (a) and (b) compared to hCEACAM1 WT (RMSD of

0.4 Å over 638 atoms and 0.6 Å over 628 atoms, respectively) (Fig. 5c, Supplementary Fig. 8b). Abrogation of the GFCC′-mediated homodimer in the context of the N97A substitution was consistent with our SEC-MALS data (Fig. 1b) and previous studies[16] demonstrating monomeric properties of the N97A mutant in solution (Fig. 1b). In fact, only two residues participated in hydrogen bond interactions less than 3.5 Å apart between two molecules of N97A in the static crystal structure: Q26 of molecule (a) and I67 of molecule (b) (Fig. 5c, Supplementary Fig. 8b). Thus, the hCEACAM1 IgV N97A crystal structure highlighted the critical properties of an asymmetric hydrogen bond network crucial towards mediating hCEACAM1 IgV dimerization.

To further confirm the monomeric state of the N97A mutant, we performed PDB-PISA (protein interfaces, surfaces and assemblies) analysis[30] and observed a complex significance score (CSS) of 0.0 for the two hCEACAM1 molecules present in the N97A structure compared to a CSS of 0.89 for the hCEACAM1 WT homodimer (Supplementary Table 4), highlighting the monomeric nature of the N97A mutant. Extension of the PDB-PISA analysis to the V39A and E99A mutants demonstrated a direct correlation with the SEC-MALS data. A CSS of 0.0 was calculated for the weak GFCC′-mediated V39A dimer (formed by molecules (c) and (d)) consistent with a monomeric species whereas a CSS score of 1.0 was calculated for molecules (a) and (b) (Supplementary Table 4) consistent with a dimer and thus suggestive of the possibility of both hCEACAM1 V39A monomers and dimers and furthermore, a rapid monomer–dimer exchange model. Similarly, we observed a CSS score of 0.63 for the E99A mutant, which demonstrated the presence of a slightly weakened GFCC′ face dimer (Supplementary Table 4). Overall, our structural studies demonstrated atomic level contributions from V39, I91, N97, and E99 within the GFCC′ face to promote hCEACAM1 homodimer formation. Further, they reveal that whereas the V39A mutant crystallized in a weak dimer

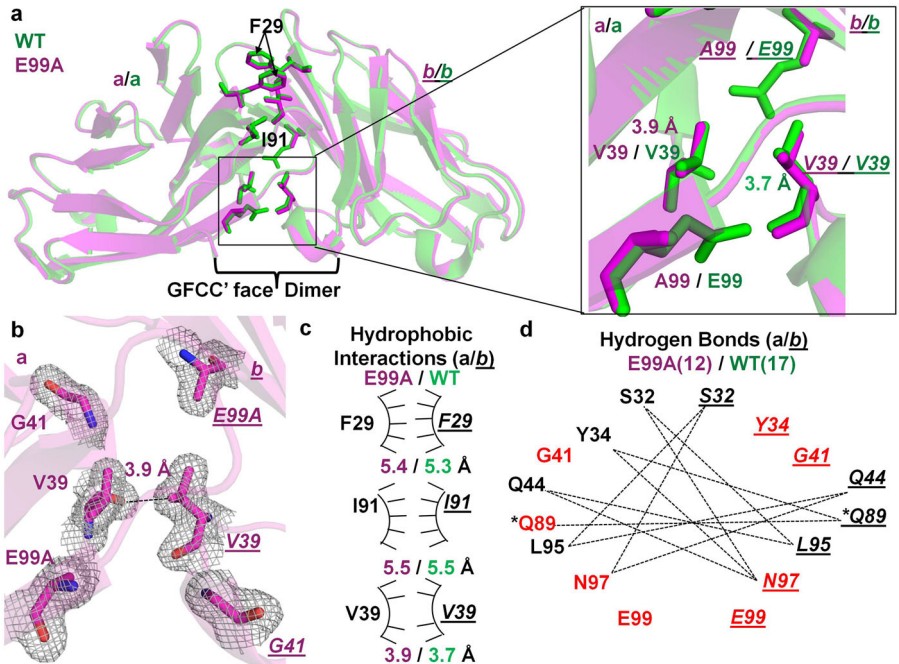

**Fig. 2 Crystal structure E99A IgV mutant of hCEACAM1. a** The ribbon diagram of the E99 mutant (magenta) and WT (green) crystal structures with molecules (a) and (b) superimposed on each other. The labeled residues of molecule (a) and molecule (b) are shown by stick representation. The inset shows the superimposition of A99 and V39 residues of the E99A mutant (magenta) on E99 and V39 residues of the WT (green), where higher distance of 3.9 Å (magenta) between the β carbons of V39 residues was observed in the E99A mutant structure compared to distance of 3.7 Å (green) between the β carbons of V39 residues of the WT. The residues of molecule (b) are shown in italics and underlined, here and throughout. **b** The stick representation of E99A, V39 and G41 residues of molecules (a) and (b) with electron densities (2Fo−Fc map at 1.0 σ level) as observed in the E99A mutant structure, which depicts loss of critical intermolecular hydrogen bonds between E99A and G41 as observed in the WT structure between the backbone amide of residue G41 with the side chain carboxyl group of residue E99. The hydrogen bond (3.9 Å) between β carbons of V39 residues as observed in the E99A mutant is shown by dashed lines. The carbon atoms in magenta, carbonyl oxygen in red, and nitrogen in blue, are colored, respectively. **c** The arc/stick representation of hydrophobic interactions by F29, I91, and V39 residues of molecules (a) and (b) as observed in the E99A crystal structure (magenta) compared to the WT (green). The hydrophobic interactions as measured by distance between β carbons of labeled residues are shown in magenta and green for E99A mutant and WT, respectively. The weaker hydrophobic interactions mediated between two V39 residues are shown by fewer pointers on the hydrophobic arc relative to WT. **d** The hydrogen bonded interactions (dashed lines) mediated by GFCC′ face labeled residues as observed in the E99A mutant structure. The lesser quantity of hydrogen bonds at GFCC′ interface were observed in the E99A mutant structure, 12 (magenta) vs 17 (green) for E99A and WT, respectively. The residues in red highlight the residues involved in the complete loss or decreased number of hydrogen bond formed in the E99A mutant structure compared to WT. The asterisk (*) indicates formation of two hydrogen bonds (shown by single dashed line) mediated by Q89 residues of molecule (a) and (b) with each other via OE1 and NE2 atoms.

configuration consistent with a transition state through effects on the CC′ loop, the N97A mutant conforms to a monomeric form of hCEACAM1.

**NMR structural studies of the hCEACAM1 dimer and N97A mutant.** To probe the role of the GFCC′ face and N97A mutation in determining hCEACAM1 dimerization characteristics in solution, we performed NMR spectroscopy studies using iso-topically labeled WT and N97A mutant IgV proteins (Figs. 6a, b, 7a, b, Supplementary Fig. 9a, b). A $^{15}$N/$^{13}$C/$^{2}$H triple-labeled hCEACAM1 protein sample (purified using our published protocol[13]) was needed to acquire the full non-uniformly sampled (NUS)[31] data set that enabled us to complete 100% of the NMR backbone assignments (excluding prolines) for the hCEACAM1 WT IgV dimer (Supplementary Fig. 9a), compared to a previously published 76% assignment[32]. The secondary structures of hCEACAM1 WT were predicted from the assigned NMR che-mical shift values using TALOS-N[33] and agree well with sec-ondary structures[8] observed in the WT homodimer (Supplementary Fig. 10a). The newly assigned residue G41 has a remarkable downfield shifted $^{15}$NH peak position in the $^{15}$N-HSQC spectrum (Supplementary Fig. 9a) which is typically associated with strong hydrogen-bond effects. Indeed, the crystal

structure of the WT homodimer demonstrates that the backbone amide of residue G41 forms an intermolecular hydrogen bond with the side chain of residue E99 (Supplementary Fig. 2a, b). Thus, Gly41 provided us with a distinctive NMR indicator for WT homodimer formation. Further, we report the overall fold of the NMR tertiary structure of WT calculated from the complete chemical shift assignments using the BMRB CS-Rosetta server[34]. The top ten lowest energy NMR predicted structures of hCEA-CAM1 WT superimpose well with the WT structure with a RMSD of 0.7 Å (over 664 atoms) and provide reliable structural descriptions of hCEACAM1 in solution (Supplementary Fig. 11).

Next, we purified $^{15}$N/$^{13}$C double-labeled N97A protein to study its behavior in solution. The $^{15}$N-HSQC spectrum of the N97A mutant showed large-scale spectral changes compared to WT (Fig. 6a) as the result of a single residue substitution. The molecular size of the N97A IgV protein was estimated by TRACT (TROSY for RotAtional Correlation Times) experiment[35] to be ~11 kD, consistent with a monomer as observed in the N97A crystal structure. We were able to complete 90% of the backbone amide resonance assignments for the N97A mutant (Supplemen-tary Fig. 9b). We also produced a $^{15}$N/$^{13}$C/$^{2}$H triple-labeled N97A protein sample which only provided limited improvement in the N97A mutant assignments (the reason will be discussed below).

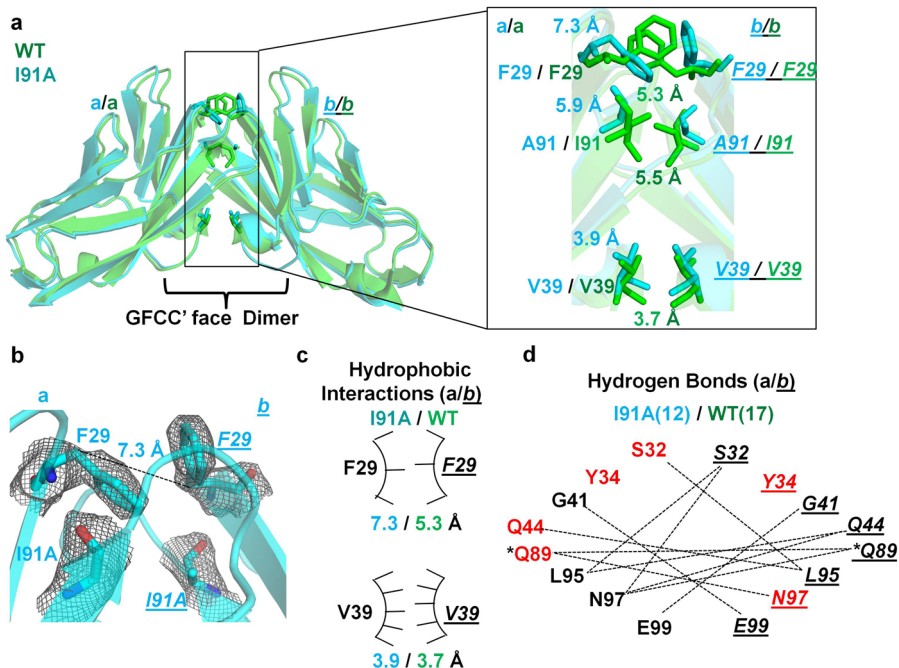

**Fig. 3 Crystal structure I91A IgV mutant of hCEACAM1. a** The ribbon diagram of the I91A mutant (cyan) and WT (green) crystal structures with molecules (a) and (b) superimposed on each other. The inset shows residues of molecule (a) and (b) by stick representation and superimposition of F29, V39, and A91 residues of the I9AA mutant (cyan) on F29, V39, and I91 residues of the WT (green), where distances between the β carbons of the labeled are shown in cyan and green, respectively for I91A mutant and WT. **b** The stick representation of I91A and F29 residues of molecules (a) and (b) with electron densities (2Fo–Fc map at 1.0 σ level) as observed in the I91A mutant structure. The hydrogen bond (7.31 Å) between β carbons of F29 residues as observed in the I91A mutant is shown by dashed lines. The carbon atoms in cyan, carbonyl oxygen in red and nitrogen in blue, are colored, respectively. **c** The arc/stick representation of hydrophobic interactions with distance between β carbons of labeled residues of molecules (a) and (b) as observed in the I91A crystal structure (cyan) compared to the WT (green). The residues of molecule b are shown in italics and underlined, respectively. The weaker hydrophobic interactions mediated between two F29 residues are shown by fewer pointers on the hydrophobic arc relative to WT. **d** The hydrogen bonded interactions (dashed lines) mediated by GFCC′ face labeled residues as observed in the I91A mutant structure. The lesser quantity of hydrogen bonds at GFCC′ interface were observed in the I91A mutant structure, 12 (cyan) vs 17 (green) for I91A and WT, respectively. The residues of molecules (a) and (b) are shown in bold and italics underlined, respectively. The residues in red indicate loss or decreased number of hydrogen bond interactions in the I91A compared to WT and asterisk (*) indicates two hydrogen bonds mediated by Q89 as described in Fig. 2d.

The overall secondary structures of the N97A mutant as predicted by the assigned NMR chemical shifts were similar to those of the WT protein (Supplementary Fig. 10b) and also consistent with secondary structures observed in the N97A mutant crystal structure. The largest backbone amide NMR chemical shift changes for the N97A mutant relative to the WT were found among residues at or near the GFCC′ face (Figs. 6b, 8a). The downfield G41 peak (indicator of an intermolecular hydrogen bond in the WT homodimer) was absent in the $^{15}$N-HSQC spectrum of N97A mutant, confirming a global conformation transition from a dimer to monomer state. Overlay of the $^{15}$N-HSQC NMR spectra obtained from N97A samples at varying concentrations (16–500 μM) revealed clear patterns of concentration-dependent chemical shift changes (Fig. 7a). The residues that shifted the most followed a distribution pattern (Fig. 7b) similar to the chemical shift differences observed between WT and N97A mutant proteins at the GFCC′ face (Fig. 6a, b), consistent with the notion that the N97A protein resides in a rapid monomer–dimer dynamic equilibrium in the fast-to-intermediate exchange regime on the NMR timescale. Fitting the peak trajectories of residues Q44 and A49 provided a rough estimate of over 1 mM affinity (dissociation constant $K_D$) for the hCEACAM1 N97A homodimer, consistent with the previous studies[16]. In contrast, the $^{15}$N-HSQC NMR of 10 μM and 1 mM WT samples did not show concentration-dependent chemical shift changes, consistent with its strong dimer association ($K_D$ = 450 nM)[16].

Although our NMR data largely support the notion that the overall structural fold of N97A in solution is highly similar to the crystal structure model, there was a region of ambiguity around the FG loop. Residues from V90 to E98A could not be readily assigned using conventional NMR sequential connectivity methods. On the other hand, there were also several NMR peaks in the $^{15}$N-HSQC spectra with weak or moderate intensities that remained unassigned because of weak NMR cross-correlation peaks. This is unlikely due to protein aggregation because there were no improvements when using a triple-labeled protein sample that was perdeuterated to reduce NMR relaxation for large proteins. It is possible that the FG loop of the N97A monomer undergoes intermediate rate exchange of multiple conformational states, likely related to the dynamic monomer–dimer equilibrium and/or additional conformational changes.

To confirm this hypothesis, $^{15}$N NMR relaxation studies of a 300 μM N97A sample were carried out by measuring the longitudinal $T_1$, transverse $T_2$(CPMG) and $T_{1\rho}$ relaxation times (Supplementary Fig. 12a). The increased $T_1/T_2$ ($R_2/R_1$) ratios for several residues in the C, C′, C″ strands, and potentially the FG loop region, were initially interpreted as reflecting local dynamic motions. However, the differences between the transverse relaxation rates $R_2$ (CPMG) and $R_{1\rho}$ derived $R_2$* (spin-lock) showed the same pattern, and clearly confirmed its origin to be from NMR chemical shift exchange processes during relaxation measurements[36] (Supplementary Fig. 12b, c). These residues

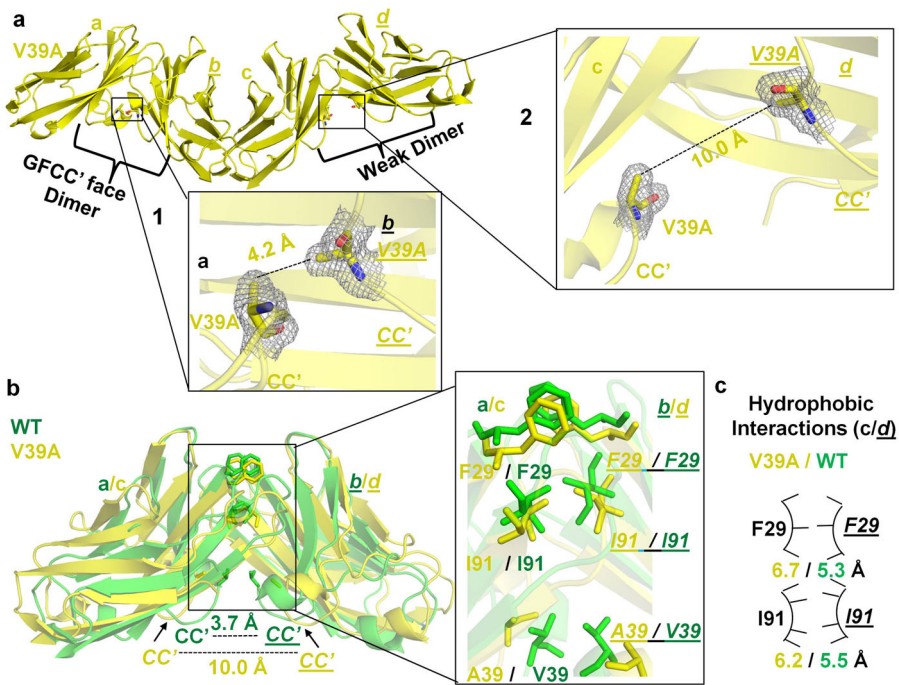

**Fig. 4 Crystal structures of the V39A IgV mutant of hCEACAM1. a** Ribbon diagram (yellow) of the molecules (a), (b), (c), and (d) as observed in the unit cell of the V39A crystal structure. The molecules (a) and (b) make a dimer that mimics a GFCC′ face dimer as observed in the hCEACAM1 (PDB code 4QXW) crystal structure (1 inset). The molecules (c) and (d) make a weak GFCC′ face dimer where the FG and specifically the CC′ loops are far apart and very few GFCC′ face residues mediate the interactions (2 inset). The insets show V39A residues of these four molecules by stick representation with electron densities (2Fo–Fc map at 1.0 σ level) and the carbon atoms in yellow, carbonyl oxygen in red, and nitrogen in blue, are colored, respectively. In the 1 inset, the distance between β carbons of V39A residues of the molecules (a), and (b) is shown by dashed line (4.2 Å), whereas the 2 inset shows increased distance between CC′ loops of weak dimer with distance of 10.0 Å between β carbons of V39A residues of the molecules (c) and (d). **b** The superimposition of the V39A mutant weak dimer (molecules c and d, colored yellow) and WT (molecules a and b colored green), whereas CC′ loops are further apart in the V39A mutant weak dimer (yellow) compared to WT (green) as measured by distance between β carbons of V39A residues of 10.0 Å in the V39A mutant weak dimer vs 3.7 Å for WT. The inset shows stick representation of the residues F29, A39 and I91 of the weak V39A dimer (yellow), which make weaker hydrophobic interactions compared to WT F29, V39, and I91 residues (green). The A39 and V39 residues of the weak V39A dimer and WT dimer are shown by yellow stick and green stick/solid arrows, representation, respectively. The residues of molecule (b) of WT (green) and molecule (d) of weak V39A dimer (yellow) are shown in italics and underlined, respectively. **c** The arc/stick representation of weaker hydrophobic interactions by F29, and I91 residues as observed between molecules (c) and (d) in the formation of weak V39A dimer of the V39A crystal structure relative to WT. The residues of molecule (d) are shown in italics and underlined, respectively. The distances of hydrophobic interactions as measured by distance between β carbons of labeled residues are shown in yellow and green for the V39A weak dimer and WT, respectively, and weaker hydrophobic interactions for V39A weak dimer are depicted by fewer pointers on the hydrophobic arc relative to WT.

exhibited relatively large $^{15}$N chemical shift differences, such that the exchange rate from the monomer–dimer equilibrium (300 μM N97A) falls within the intermediate time scale on the order of milliseconds. These results underscore the important effects that the N97A mutation has on disrupting the extremely stable dimeric form of hCEACAM1 WT, and strongly shifting the dynamic monomer–dimer exchange towards a monomeric form.

**Conformation and thermal motion analysis of hCEACAM1 WT IgV and GFCC′ face variants**. To investigate hCEACAM1 IgV domain conformational flexibility, and specifically at the CC′ and FG loops, we compared the structural fold and dynamic mobility of the hCEACAM1 IgV domain from previously published crystal structures of hCEACAM1 WT (PDB codes 4QXW, 2GK2), hCEACAM1 WT-HopQ complex (PDB code 6AW2)[8,12,16] and our crystal structures of the various hCEACAM1 alanine substitutions. Although structural superimposition of a hCEACAM1 molecule from the WT homodimer (PDB code 4QXW) revealed a similar GFCC′ global fold with RMSD of 0.6 Å (over 724 atoms) and 0.5 Å (over 598 atoms) with the hCEACAM1 WT structure (PDB code 2GK2) and hCEACAM1-HopQ

complex (PDB code 6AW2), respectively, considerable conformational differences were observed in the FG loops (Fig. 8b, Supplementary Fig. 13). Assessment of the dynamic mobility (as measured by the Debye-Waller factor or B factor) of each hCEACAM1 IgV molecule revealed an average B factor of ~23 Å$^2$ for a hCEACAM1 WT dimer (PDB code 4QXW) with similar B factor values observed across the GFCC′ face (Table 1). In comparison, we observed average B factors of ~28/16/31 Å$^2$ for V39A/N97A/E99A, respectively, in the refined crystal structures (Table 1) that negatively correlated with their respective calculated melting temperatures (Fig. 1a), where increased B factor was associated with reduced thermal stability. While the crystal structure of the N97A mutant showed the lowest B factor value of 16 Å$^2$, a closer examination revealed higher B factor profiles of the CC′, EF, and FG loop residues (Fig. 8c). Packing constraints combined with the time and space averaging of a crystal structure determination limit the interpretation of a protein's behavior in solution[37], however, the higher B factor of the CC′ and FG loop observed for the N97A structure relates well with the multiple conformational states of CC′ and FG loop residues observed in our NMR experiments.

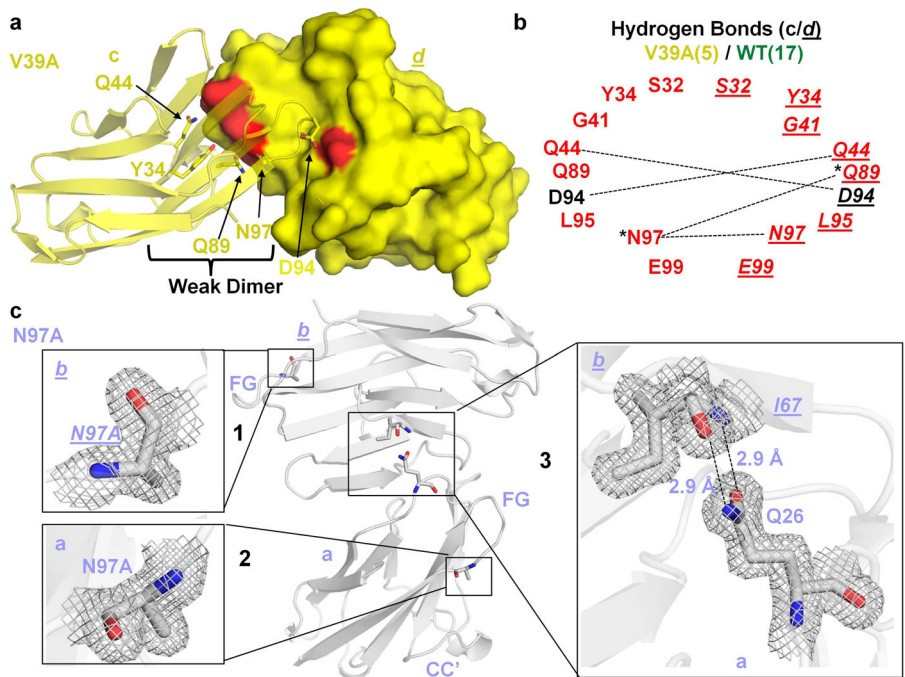

**Fig. 5 V39A weak dimer hydrogen bonded interactions and crystal structure of monomeric N97A IgV mutant. a** The molecules (c) and (d) that mediate formation of a weak GFCC′ face V39A dimer are shown in ribbon and surface representation in yellow, respectively. The residues Y34, Q44, Q89, D94, and N97, which mediate the hydrogen bonded interactions, are shown in stick representation for molecule (c) and surface bright and light red representations for molecule (d). The carbon atoms in yellow, carbonyl oxygen in red and nitrogen in blue, are colored, respectively. **b** The fewer hydrogen bonded interactions that result in the weak V39A dimer formation between molecules (c) and (d) residues of the V39A mutant crystal structure are shown by dashed lines. The residues of molecule (d) are shown in italics and underlined, respectively. The lesser quantity of hydrogen bonds at GFCC′ interface were observed in the V39A weak dimer, 5 (yellow) vs 17 (green) for V39A dimer and WT, respectively. The residues in red indicate loss or decreased number of hydrogen bond interactions in the V39A weak dimer compared to WT. The asterisk (*) indicates the formation of two hydrogen bonds (shown by single dashed line) mediated between N97 residue of molecule (c) and Q89 residue of molecule (d). **c** The crystal structure of N97A mutant with two monomeric molecules (a, bottom) and (b, top) shown by ribbon diagram colored silver white. The carbon atoms in silver white, carbonyl oxygen in red and nitrogen in blue, are colored, respectively. The insets 1 and 2 shows stick representation of N97A residues of both molecules with electron density (2Fo–Fc map at 1.0 σ level) and this mutation leads to abrogation of GFCC′ face dimer in the N97A crystal structure. The residues of molecule (b) are shown in italics and underlined, respectively. The CC′ and FG loops are labeled and very limited interface contact between two N97A molecules through ABED face is shown in inset 3, where Q26 of molecule (a) and I67 of molecule (b) participates in hydrogen bonded interaction. The two hydrogen bonds of 2.9 Å and 2.9 Å between the aforementioned residues are shown by dashed lines. The residues are shown by stick representation with electron density (2Fo–Fc map at 1.0 σ level).

## Discussion

We performed biophysical, high resolution crystallography, and NMR studies to determine the basis for CEACAM1 monomer–dimer exchange at an atomic level. Our initial biophysical studies confirmed the previously described dimer-disruptive property of the N97A mutant in solution[16] and demonstrated that the hCEACAM IgV domain exchanges between monomer and dimer forms. Furthermore, GFCC′ face-targeted mutational studies using V39A, I91A, N97A, and E99A mutant proteins provided an experimental opportunity to shift the monomer–dimer equilibrium towards each species, highlighting the unique thermally stable N97A monomeric mutant. Crystal structures provided static snapshots that included a possible monomer–dimer transition state (V39A) and complete monomeric state (N97A), while NMR studies demonstrated the dynamic properties of the CC′ and FG loops of the N97A mutant in solution. These studies focus attention on the important role of the GFCC′ face in determining hCEACAM1 monomer–dimer equilibrium. The weakening of GFCC′ face-mediated dimer association was manifested by disruption of many CC′ and FG loop residues interactions observed in the V39A mutant and complete loss of CC′ and FG loop residues interactions in the N97A mutant, suggesting that the GFCC′ face may sequentially "unzip" and "rezip" in transitioning between a dimeric and

monomeric state. Consistent with this hypothesis, our NMR studies of the N97A mutant revealed that monomer–dimer exchange involved residues within the GFCC′ face, including V39, Y48, Q89, and A100 (Fig. 7a).

An important contribution of our study is the full backbone assignment of the NMR spectra of wildtype hCEACAM1 IgV and near complete (90%) assignment of the NMR spectra of the hCEACAM1 IgV N97A mutant. The unassigned residues, largely in the FG loop that undergoes the most drastic conformational change in the context of the N97A substitution, might be a consequence of exceptionally large $^{15}$NH chemical shift differences leading to complete exchange line-broadening resulting in extremely weak and/or missing resonance peaks. The higher conformational flexibility and B factor of FG loops residues as described before for WT, N97A, and the hCEACAM1-HopQ complex (Fig. 8b–c) support this local structural malleability which could facilitate hCEACAM1 IgV homodimer formation as well as the formation of complexes with many other proteins, including TIM-3[8,13,18], HopQ[16] and other microbial pathogens[10].

Our studies are consistent with a model wherein the hCEACAM1 IgV GFCC′ face represents the primary face involved in homodimer and heterodimer formation (Fig. 9). When hCEACAM1 transitions from a dimer to monomer, possibly through disruption of CC′ loop interactions (as observed in the V39A

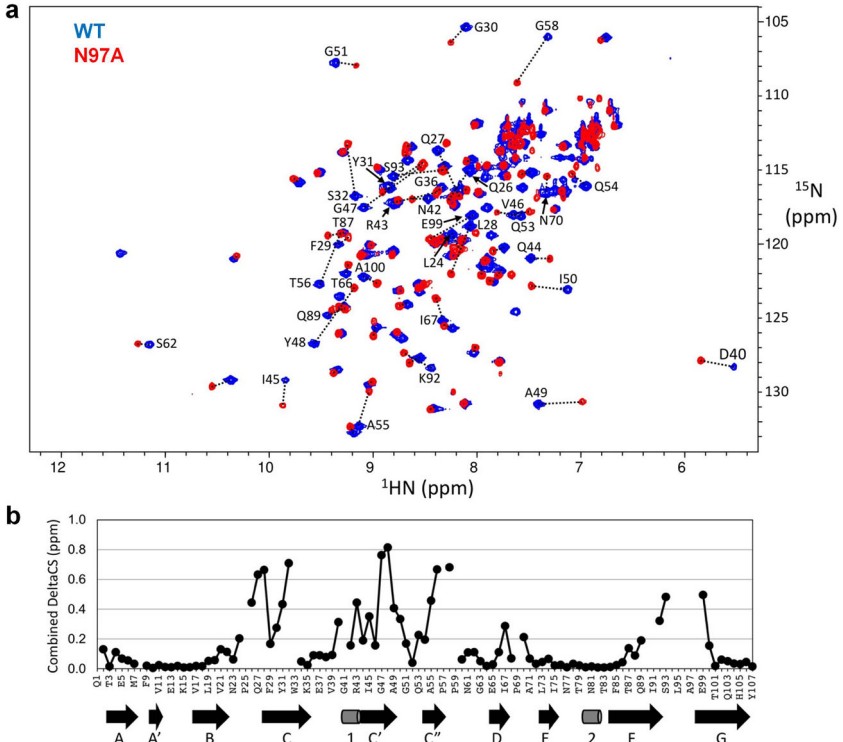

**Fig. 6 15N-HSQC spectra of WT and N97A mutant hCEACAM1 IgV protein. a** Overlaid 15N-HSQC spectra of WT (blue) and N97A mutant (red) hCEACAM1 IgV protein. The corresponding assigned residues with major peak shifts were indicated and connected by dotted lines. **b** Combined 1HN and 15N chemical shift changes, sqrt(δcsH2 + (δcsN/5)2), between WT and N97A mutant hCEACAM1 IgV are shown in comparison with the secondary structure elements from the X-ray structure of WT hCEACAM1 depicted below.

weak dimer GFCC′ face crystal structure), there is higher thermal motion and dynamic conformations within the CC′ and FG loops that accommodate a monomeric hCEACAM1 species that is readily amenable to participating in both homophilic and heterophilic interactions. Homophilic interactions mediated by the GFCC′ face of a hCEACAM1 monomer with a neighboring hCEACAM1 monomer either in *cis* or *trans* return the destabilized hCEACAM1 monomers to a more thermally stable dimeric form. The GFCC′ face-stabilized homodimer could subsequently participate in higher order oligomer formation, possibly through minor interactions mediated though the ABED face (Fig. 9). Notably an ABED-mediated homodimerization interface has also been suggested in SPR binding studies[11] and described in a crystal structure of non-glycosylated hCEACAM1 WT IgV (PDB code 2GK2)[12]. Although the ABED surface contains three glycosylation modification sites, an attractive possible contribution of the ABED face is to serve as a flexible secondary homo-oligomerization site that achieves relevance following *trans* GFCC′-initiated homo or heterodimerization by propagating surface CEACAM1 clustering and downstream signal activation. Higher order oligomerization enables interactions between CEACAM1 ITIM-containing cytoplasmic tails that impart inhibitory signals through association with *Src*-homology domain-containing phosphatases. Indeed, many functional studies have also demonstrated that the propensity of CEACAM1 to form higher order oligomers may be initiated by formation of monomers through transmembrane or cytoplasmic tail interactions with calcium followed by GFCC′ face interactions[11,38]. Interestingly, up to fifty percent of CEACAM1 on the cell surface of CEACAM1 transfected cells has been predicted to be in a monomeric state with the remainder existing as homodimers consistent with a monomer–dimer equilibrium in the physiologic function of this important cell surface protein[10]. Thus, our

structural model extends our understanding of the hCEACAM1 monomer–dimer equilibrium and provides a structural rationale for oligomerization-mediated activities.

Our findings also help to understand how the dynamic nature of the hCEACAM1 GFCC′ face facilitates its binding with various other host ligands, such as hTIM-3 hCEACAM5, PD1, and numerous pathogen proteins (Supplementary Fig. 14). hTIM-3, in particular, is an important immunoregulatory protein that possesses an N-terminal IgV domain with high structural similarity to hCEACAM1. Using various cellular, biochemical and biophysical methods[13,18,39–41], we and others have demonstrated a conserved role of the GFCC′ faces of hCEACAM1 and hTIM-3 in hCEACAM1-hTIM-3 heterodimer complex formation ($K_D$ of ~2–3 μM) (Supplementary Fig. 14). Despite these findings, a recent paper[42] suggested a lack of appreciable binding of hCEACAM1 by hTIM-3. This is surprising and likely due to a number of factors, including the use of incompletely characterized Fc fusion proteins that do not take into account the monomer–dimer equilibrium or structural state of the proteins used or consideration of the relative affinities ($K_D$) of the hCEACAM1 homodimer (~450 nM) and hCEACAM-hTIM-3 heterodimer (~2–3 μM). In the ELISA studies for example a single concentration of immunoglobulin fusion proteins was used in the nanomolar range without the control of the calcium level, which naturally shifts the interaction towards detection of higher affinity hCEACAM1 homophilic binding rather than lower affinity hCEACAM1-hTIM-3 interactions. Further, there is an absence of titration experiments in the micromolar range to probe the binding and there are confounding results that show that the strongest ligand binding to the hCEACAM1-Ig used was with galectin-9 in the absence of galectin-9 binding to hTIM-3. Importantly, galectin-9 has never been described as a ligand for hCEACAM1 and numerous groups have unambiguously

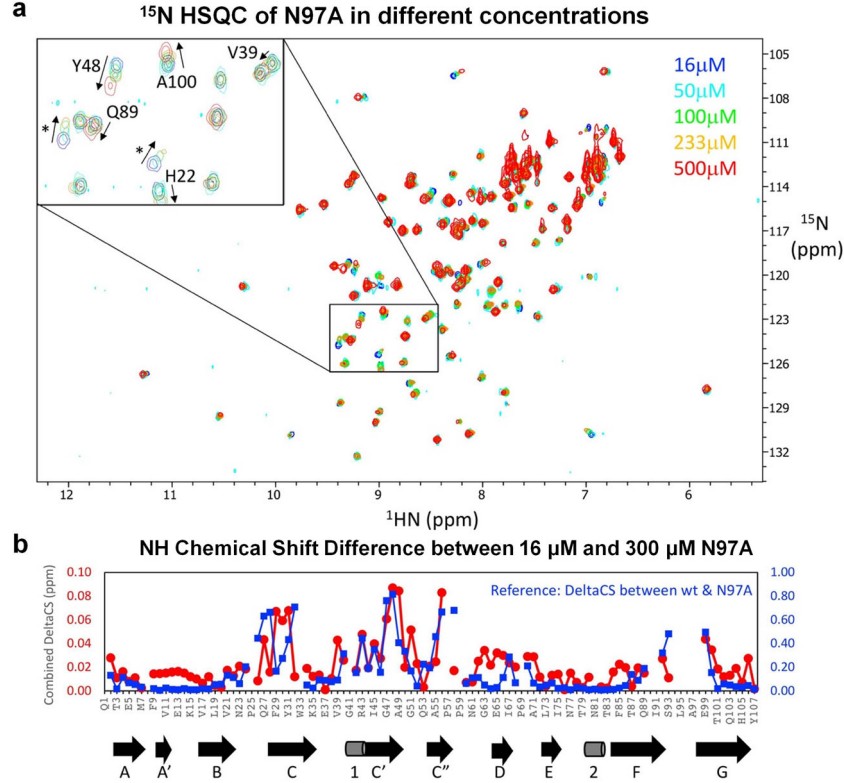

**Fig. 7 $^{15}$N-HSQC spectra and combined chemical shift changes of N97A mutant at different concentrations. a** Overlaid $^{15}$N-HSQC spectra of 16 μM (blue), 50 μM (cyan), 100 μM (green), 233 μM (orange) and 500 μM (red) N97A mutant. The inset shows an enlarged view of a central spectral region containing several assigned residues and unassigned residues (marked with "*"). **b** The relative combined $^{1}$HN and $^{15}$N chemical shift changes of assigned $^{15}$NH peaks between a 16 μM and a 300 μM N97A mutant protein sample (red circles), in comparison with the relative combined $^{1}$HN and $^{15}$N chemical shift changes of $^{15}$NH peaks between the WT and N97A mutant proteins (blue squares). The secondary structure elements from the X-ray structure of N97A protein are depicted below for comparison.

identified an interaction between hTIM-3 and galectin-9[39,43–45], raising important concerns about the interpretation of these and other results contained therein[42].

The homo-oligomeric and hetero-oligomeric properties of CEACAM1, especially with regards to TIM-3, PD1, and microbial ligands, carry important therapeutic potential making our understanding of the CEACAM1 monomer to dimer transition and associated receptivity of the CEACAM1 monomer of great importance. A number of groups[18,39–41] have recently observed that selective targeting of the GFCC′ faces of either hCEACAM1 (e.g., with the 5F4 monoclonal antibody or hTIM-3 peptides) or hTIM-3 (e.g., with polyclonal antibodies or monoclonal antibodies including 2E2 or M6903) can disrupt the formation of hCEACAM1 homodimers and complexes with hTIM-3, respectively, using therapeutic agents that exceed the natural homo-dimeric and heterodimeric affinities (Supplementary Fig. 14). Consistent with these findings, hCEACAM1 binding by *Neisseria sp.* OPA proteins or *E. coli* Afa/Dr adhesins and the recent hCEACAM1-HopQ interaction studies support the critical importance of the GFCC′ face and monomer–dimer equilibrium. In the case of HopQ, *H. pylori* has developed opportunistic mechanisms to specifically target the GFCC′ face on CEACAM1 IgV and interfere with CEACAM1 homodimerization ($K_D$ ~450 nM) through enhanced heterophilic interactions ($K_D$ ~23–279 nM). The crystal structure of the HopQ-hCEACAM1 IgV complex illuminates the ability of HopQ to achieve the formation of a high affinity heterodimer through its interaction with the same residues (V39, I91, N97) fundamental for hCEACAM1

homophilic interactions. Thus, these recent findings further extend the role of GFCC′ face residues in interactions with many different heterophilic ligands in immune-regulation and immune-evasion that are exploited by neoplastic cells and microbial pathogens (Supplementary Fig. 14).

In summary, our biophysical and structural studies by crystallography and in solution by NMR support a model wherein the GFCC′ face is highly dynamic and seeks thermal and energetic stability through the formation of dimers (either homodimers or heterodimers) that lock in a structurally favorable state. As such, CEACAM1 prefers to be in a dimeric state that specifically stabilizes the CC′ and FG loops, thus making the GFCC′ face the major interaction site responsible for homodimer and hetero-dimer complex formation. A caveat of these studies is that they were performed with unglycosylated proteins that may affect the ABED face; however, studies reporting a role of glycosylation in disrupting homodimerization have been corrected[32,46]. That said, given the location of the carbohydrate side-chain modifications of CEACAM1 along the ABED face, the mutational analyses performed here and its implications still have substantial physiologic merit. In addition to furthering our understanding of the structural mechanisms that underlie the formation of CEACAM1 monomers that are amenable to varying interactions with another CEACAM1 molecule or its potential heterophilic partners, our high-resolution structural studies and hCEACAM1 dynamic monomer–dimer equilibrium model provide a beneficial foundation towards therapeutic targeting of hCEACAM1 interactions with its various ligands.

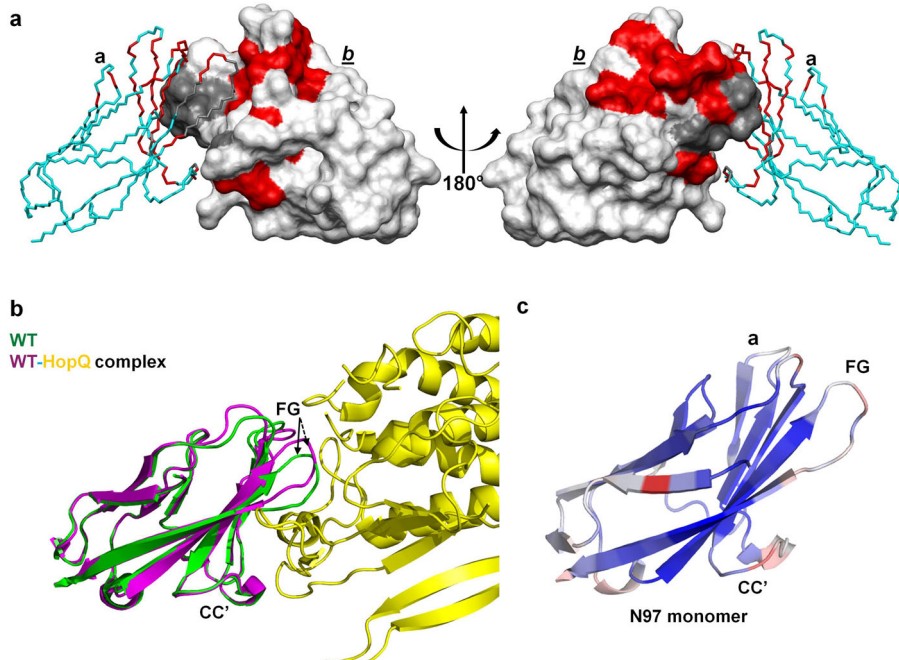

**Fig. 8 Mapping of NMR chemical shift changes of N97A mutant rendered on WT structure and conformational flexibility of the GFCC′ face. a** Front (left) and back (right) views of the crystal structure of the hCEACAM1 WT IgV dimer (PDB code 4QXW) with one unit shown in molecular surface representation (white) and another in alpha carbon traces (cyan). The residues in N97A mutant hCEACAM1 IgV with combined ${}^{1}$HN and ${}^{15}$N NMR chemical shift changes larger than 0.2 ppm are colored in red, while residues with missing assignments presumably due to dynamic conformation exchanges are colored in gray. **b** Structural alignment of a hCEACAM1 molecule of WT homodimer in green (PDB code 4QXW) with a hCEACAM1 molecule of WT-HopQ complex crystal structure in magenta (PDB code 6AW2). For clarity, only half of the HopQ molecule (yellow) is shown. CC′ and FG loops are labeled, and the conformational differences of the FG loop are depicted by an arrow. **c** The ribbon diagram of the molecule (a) of N97A crystal structure with B-factor assignment. The loops, α helices, and β strands are colored based on B-factor range (blue-white-red, where blue minimum = 10, red maximum = 20).

## Methods

**Protein expression and purification**. The hCEACAM1 WT IgV and mutant (V39A, I91A, N97A, E99A) proteins were expressed and purified using our published protocols[13].

**Differential scanning fluorimetry**. Differential scanning fluorimetry was performed using a QuantStudio 6 (Life Technologies) RT-PCR instrument with the excitation and emission wavelengths set to 587 and 607 nm, respectively. Assay buffer was 10 mM HEPES pH 7.4, 150 mM NaCl. For thermal stability measurements, the temperature scan rate was fixed at 1 °C/min. Protein concentration was uniform at 25 μM among the hCEACAM1 WT and mutant samples and SYPRO orange (Invitrogen) concentration was consistent at 5×. The temperature range spanned 20 °C to 95 °C. Data collection was performed by Quant Studio Real-Time PCR Software (Life Technologies) on triplicate samples and analyzed by Protein Thermal Shift Software v1.4 (ThermoFisher). Melting point temperature ($T_m$) was calculated for each protein samples through computation of a temperature derivative for each respective melting curve that was then processed with a peak fitting algorithm, applying a sigmoidal baseline and fitting the peak to determine the $T_m$ and its standard error.

**Size-exclusion chromatography with multi-angle light scattering**. Purified samples of WT and mutant (V39A, I91A, N97A, E99A) hCEACAM1 IgV were evaluated for size and monodispersity by analytical size exclusion chromatography and multi-angle light scattering (SEC-MALS). Samples of hCEACAM1 at 100 μM were injected onto a TSK-gel Bioassist G4SWxl (Tosoh) SEC column, equilibrated with 10 mM HEPES pH 7.4, 150 mM. The SEC column was coupled to a static 18-angle light scattering detector (DAWN HELEOS-II) and a refractive index detector (Optilab T-rEX) (Wyatt Technology, Goleta, CA). Data were collected at a flow rate of 0.5 mL/min. Retention time, molecular weight, and polydispersity index (PDI) were calculated in ASTRA (Wyatt).

**Crystallization, data collection, and structure determination**. Purified mutant (V39A, I91A, N97A, E99A) proteins were concentrated to 10 mg/ml (exceeding 800 μM concentration) and preliminary crystallization screens were performed using Index HT (Hampton Research) and PEGRx (Hampton Research) at 4 °C and room temperature. The conditions with promising hits were later optimized with

an additive screen (Hampton Research) and X-ray diffraction quality crystals were obtained under crystallization conditions of 0.005 M cobalt(II) chloride hexahydrate, 0.005 M nickel(II) chloride hexahydrate, 0.005 M cadmium chloride hydrate, 0.005 M magnesium chloride hexahydrate with 12% w/v polyethylene glycol 3,350 in 0.1 M HEPES pH 7.5 buffer (V39A mutant condition), 54% Tascimate with 0.5 % n-Octyl-β-D-glucoside pH 8.0 (I91A mutant condition), 6% Tascimate with 25% polyethylene glycol 400 in 0.1 M MES monohydrate pH 6.0 (N97A mutant condition), and 41% Tascimate with 0.5 % n-octyl-β-D-glucoside pH 8.0 (E99A mutant condition), respectively. For X-ray data collection, crystals were cryoprotected in mother liquor of crystallization condition with approximate concentration of 12% glycerol and 7% ethylene glycol. X-ray data for V39A, I91A, and E99A mutant protein crystals were collected at the 21-ID-F/G, LS-CAT beamlines at the Advanced Photon Source (APS; Argonne, IL, USA) and at the 17-ID beamlines, National Synchrotron Light Source-II (NSLS-II, Upton, NY, USA) for the N97A mutant protein crystals. The diffraction data for each of the mutant (V39A, I91A, N97A, E99A) protein crystals were processed with iMosflm and the CCP4 suite of software[47,48], HKL2000[49], and FastDP[50] in-house and at the beamlines (APS, Argonne, IL; NSLS-II, Upton, NY). The structure of the each mutant was determined by molecular replacement with MolRep[48] using a polyalanine model of our hCEACAM1 WT crystal structure (PDB code 4QXW) and many rounds of structure refinement were done with simultaneous model building using Refmac[51] and COOT[52]. The Fo–Fc map at 3.0 σ level (derived from the initial model) showed considerable positive Fo–Fc map density where residue A39 in the V39A refinement model, residues A91 in the I91A refinement model, residue A97 in the N97 refinement model, and residues A99 in the E99A refinement model were not fitted, respectively (Supplementary Fig. 15a–d).

The I91A and E99A mutants were crystallized in a tetragonal space group similar to hCEACAM WT IgV (PDB code 4QXW), however, with varied unit cell constants (Table 1). In contrast, the V39A and N97A hCEACAM1 IgV mutants were crystallized in unique trigonal and c-centered orthorhombic space groups, respectively. The crystallographic twining and the considerable metal electron density near the FG loop were observed in the V39A protein during the data processing and the structure refinement. The V39I mutant protein crystallization condition from Index HT screen (Hampton Research) has 0.005 M cobalt(II) chloride hexahydrate, 0.005 M nickel(II) chloride hexahydrate, 0.005 M cadmium chloride hydrate, and 0.005 M magnesium chloride hexahydrate. Using intensity based twin refinement in Refmac and looking at the interacting residues H105 and V106 around the observed metal density, we modeled the final crystal structure

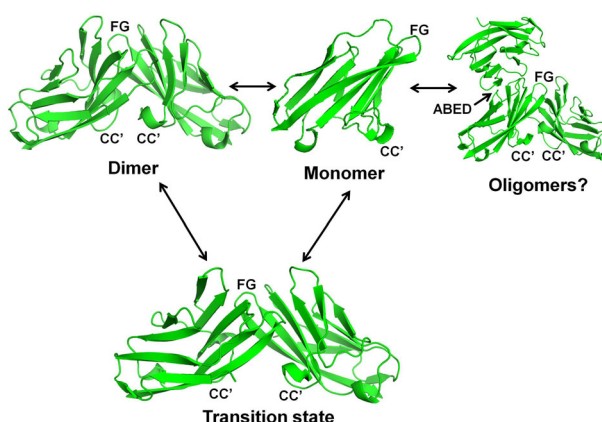

**Fig. 9 Dynamic human CEACAM1 monomer–dimer equilibrium.**
Monomer, dimer, transition and potential oligomeric states of hCEACAM1 IgV as observed in various crystal structures. The ribbon diagram of all states of hCEACAM1 are shown with labeled CC′ and FG loops. The dimer state of hCEACAM1 was observed in WT crystal structure (Supplementary Fig. 2a–d) and mediates dimer formation through GFCC′ face. The weak V39A dimer as observed in the crystal structure and described in Fig. 4a–c results from weaker hydrogen bonded and hydrophobic interactions at the GFCC′ face and mimics a transition state. The N97A mutation results in the complete abrogation of GFCC′ face dimer as described in the Fig. 5c. The monomeric state of N97A mutant as observed in the crystal structure is further confirmed by NMR studies (Fig. 6a, b). Further, hCEACAM1 could form higher order soluble oligomers over time which could possibly occur through a major association of the GFCC′ face's two monomers together with minor interactions through a flexible ABED face as observed in WT (PDB code 2GK2), V39A (Supplementary Fig. 7), and N97A crystal (Fig. 5c) structures where minor points of contact through an ABED face were observed as shown in the model by superimposing the WT dimer (4QXW) and N97A crystal structure.

with bound $Ni^{++}$ (Nickel) for the molecule (a) or (c) and their two symmetry mates 0100–100 (S1) and 02000000 (S2) with the strategy that showed optimal R/Rfree and revealed hexadentate interactions of $Ni^{++}$ with three His105 sidechains and three carbonyl groups of Val106 residues (Supplementary Fig. 6). The final crystal structure of the V39A mutant was determined in the P3 space group with Rwork/Rfree of 14.5%/18.6% at 1.9 Å resolution. For other mutants, the crystal structure of the I91A mutant was solved in the P4212 space group with Rwork/Rfree of 22.1%/25.8 at 3.1 Å resolution, that of the N97A mutant in the C2221 space group with Rwork/Rfree of 19.2%/24.0% at 1.8 Å resolution, and the E99A mutant in the P4212 space group with Rwork/Rfree of 18.9%/22.3 at 1.9 Å resolution, respectively. In order to further validate observed 2Fo–Fc electron density of each mutant at the mutation site (Supplementary Fig. 15e–h), the final refined model of each mutant was changed to the original residue as present in the hCEACAM1 IgV WT (valine for the V39A refined model, isoleucine for the I91A refined model, asparagine for the N97A refined model, and glutamic acid for the E99A refined model) and an additional cycle of refinement was performed. The negative difference density in the Fo–Fc map at 3.0 σ level for each mutant at the reverse mutation site was observed and validated structures of V39A, I91A, N97A, and E99A mutants (Supplementary Fig. 15i–l). All the figures, B factor calculation and conformational mapping were carried out using PyMOL (DeLano Scientific) and sequence alignments of hCEACAM1 were done using Clustal Omega[53].

**Nuclear magnetic resonance studies**. $^{15}N/^{13}C$ double-labeled WT and N97A hCEACAM1 IgV proteins were expressed from *E. coli* in M9 minimal medium containing $^{15}NH_4Cl$ and $^{13}C$-gluocse as the sole nitrogen and carbon sources and purified as previously reported[13]. $^{15}N/^{13}C/^2H$ triple-labeled WT and N97A mutant IgV proteins were expressed similarly except in $D_2O$ instead of $H_2O$. Non-uniformly sampled (NUS) triple resonance experiments using WT $^{15}N/^{13}C/^2H$-hCEACAM1 IgV (0.2 mM) in 10 mM HEPES, 50 mM NaCl, pH 7.0 with 10% $D_2O$, were performed at 25 °C on a 700 MHz Agilent DD2 spectrometer equipped with a cryogenic probe. The data were processed using NMRPipe[54] and Iterative Soft Thresholding reconstruction approach (istHMS)[55] and analyzed by CARA[56]. Backbone assignment experiments for N97A mutant hCEACAM1 IgV were performed with a 0.3 mM $^{15}N/^{13}C$ double-labeled protein sample under the same condition using the same methods as described above. Secondary structure predictions based on assigned chemical shifts (H, HN, CO, CA, and CB) were obtained

using the TALOS-N software[33]. The NMR structure models of hCEACAM1 WT were predicted from the assigned chemical shifts by CS-Rosetta[33].

The 1D $^{15}N$ TRACT[35] experiments were carried out using $^{15}N/^{13}C$ double-labeled N97A mutant hCEACAM1 IgV protein sample at 25 °C on a 500 MHz Bruker Avance III HD spectrometer. Series of relaxation delays (in increments of 50 ms) up to 250 ms were used for pro TROSY measurements; and delays (in increments of 20 ms) up to 100 ms were used for anti TROSY measurements. Data were processed and analyzed using the Bruker Topspin program. The difference between the pro and anti TROSY relaxation rate is 12.6 s$^{-1}$, which corresponds to a correlation time of 6.5 ns and estimated molecular weight of 11 kDa.

NMR relaxation experiments were carried out using a 0.3 mM $^{15}N/^{13}C$ double-labeled N97A mutant hCEACAM1 IgV protein sample at 25 °C on a 700 MHz Agilent DD2 spectrometer equipped with a cryogenic probe. The $T_1$ relaxation times were determined using antiphase inversion recovery delays of 10 ms, 250 ms, 500 ms, 750 ms and 1 s. The $T_2$ relaxation times were determined using Carr-Pursell-Meiboom-Gill (CPMG) pulse train with τ value of 625 μs, and delays of 10 ms, 30 ms, 50 ms, 90 ms, and 150 ms. The $T_{1rho}$ relaxation times were determined with spin-locking field strength of 1.75 kHz, and delays of 10 ms, 30 ms, 50 ms, 90 ms, and 130 ms. Data were processed using NMRPipe and analyzed by CARA. The errors of the relaxation times were estimated from fitting routines. The NMR chemical exchange process for some of the N97A residues were not sufficiently suppressed by the $T_2$ CPMG pulse train (with τ = 625 μs), versus the more efficient rotating frame $T_{1\rho}$ spin-locking, resulting in artificially higher $R_2/R_1$ and $R_2/R_{1\rho}$ ratios. This exchange is more accurately described by the relaxation rate difference between $R_2$(CPMG) and $R_2$*(spin-lock), which is derived from $R_{1\rho}$ according to $R_{1\rho} = R_1 \cos2\theta + R_2^* \sin2\theta$, (where θ is the tilt angle of the effective spin-locking field in the rotating frame due to off-resonance effect).

**PDB PISA validation**. The PDB PISA (proteins, interfaces, structures and assemblies) computes a complex significance score (CSS) of macromolecular complex formation by accounting for the energy of complex formation, interface area, number of bonds formed, and hydrophobicity variables with solvation energy gain[30]. The PDB PISA validation was done using the final refined atomic coordinate file of each mutant (V39A, I91A, N97A, E99A) and complex significance scores (CSS) with residue level interactions details were determined[30]. The hydrogen bond interactions and CSS scores as observed in the crystal structures of the WT, V39A, I91A, N97A, E99A mutants are shown in Supplementary Tables 4–12, respectively.

**Statistics and reproducibility**. The X-ray data and structure refinement statistics for the V39A, I91A, N97A, E99A mutant crystal structures are shown in Table 1. Differential scanning fluorimetry studies were performed on triplicate samples (*n* = 3) of each hCEACAM1 IgV mutant representing independent biological samples as depicted in Fig. 1. The mean value melting temperature is shown in bar graph form. Error bars represent standard deviations and reflect high reproducibility of the triplicate samples.

## Data availability
The atomic coordinates and structure factors were deposited with RCSB accession code 6XNO (E99A), 6XNT (I91A), 6XNW (V39A), 6XO1 (N97A), respectively. The assigned NMR chemical shifts have been deposited in the Biological Magnetic Resonance Bank database (BMRB ID 50368 for hCEACAM1 IgV WT dimer and BMRB ID 50366 for N97A mutant hCEACAM1 IgV monomer). Source data underlying plots shown in figures are provided in Supplementary Data 1. All relevant data are available upon request.

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

## Acknowledgements

The authors thank all the beamline scientists and the support staff of LS-CAT beamlines 21-ID-F and21-ID-G (APS; Argonne, Illinois, USA) and NSLS-II 17-ID-1 and 17-ID-2 beamlines (Upton, NY, USA) for X-ray data collection, HMS EQNMR (Boston, MA, USA) and DFCI NMR (Boston, MA, USA) facilities for NMR data collection, and HMS Center for Molecular Interactions (Boston, MA) for DSF and SEC-MALS data collection. We acknowledge BMRB CS-Rosetta server for the NMR structure prediction (https://csrosetta.bmrb.wisc.edu/csrosetta/submit). We are thankful to M. Pyzik and A. Riar for helpful discussions. This work was supported by the NIH Grant 5R01DK051362-21 and the High Pointe Foundation to R.S.B., and 5P01AI073748-09 to V.K.K.

## Author contributions

A.K.G., W.M.K., Y.- H.H., and Z.-Y.J.S. performed biophysical characterization, X-ray crystallography and NMR experiments. W.M.K. performed DSF and SEC-MALS experiments. A.K.G. performed X-ray crystallization, structure determination and conformational analysis of hCEACAM1 mutants. Z.-Y.J.S. and G.W. planned NMR experiments. Z.-Y.J.S. carried out NMR experiments and assigned NMR spectra of the hCEACAM1 IgV dimer and N97A monomer. Y.- H.H. and Y.K. designed expression constructs and performed similarity analyses of hCEACAM family members. D.A.B. and E.J.S. established purifications and refolding protocol and conceptualization of conformational analysis. G.P. provided structural expertise in crystallization, refinement, B factor analysis and structural comparison. R.S.B. and V.K.K. established this hCEACAM1 project, and together with A.K.G., G.P., W.M.K., Z.-Y.J.S., Y.-H.H., and Y.K. wrote the manuscript. R.S.B. and A.K.G are the senior authors of this paper.

## Competing interests

The authors declare the following competing interests. R.S.B. has several issued and pending patents describing potential therapeutic strategies for regulating CEACAM1.
