## [Peer Review File · Communications Biology]

Reviewers' comments:

Reviewer #1 (Remarks to the Author):

This manuscript merges biophysical and structural biology methods to study single amino acid variants of hCEACAM1 to probe monomer-dimer exchange. The integration of the various techniques is a strength of the manuscript and the NMR studies are particularly compelling. Some aspects require further explanation/consideration, as described in detail below. Overall, the study provides new fundamental information on the impact of interactions at the hCEACAM1 dimerization interface.

Figure 1: Data supporting a single thermal denaturation temperature for each hCEACAM1 IgV variant is not included and probably should be based on text (p. 7, line 157). The claim that decreasing melting temperature correlates with decreasing homodimerization (p. 8, line 160) is similarly not supported. Rather as noted by the authors, the presented data (Fig. 1B) conflicts with the statement; namely, N97A. In addition, the displayed data in Fig. 1B doesn't show a single peak for each protein variant (line 171) – consider for example, I91A which clearly shows two peaks and E99A which can be fitted to two peaks.

Figure 2: In comparisons of the crystal structures more clarity is needed. For example does “weaker hydrogen bonded and hydrophobic interactions” (line 192-3) mean lesser? It's also not possible to see the change in interactions from the figs displayed. Perhaps zoomed in regions are needed along with corresponding electron density maps. The interactions maps (Supp Fig 2B/D for example) would be better as side-by-side comparisons and with corresponding structures/density maps. Referred to interactions/amino acids are in multiple figs not apparent, such as F29 in Fig. 2B (line 202) and hydrogen bonds in Fig 2D (line 207-210). There's no figure to confidently evaluate the described increase in distance for V39 (line 212). Another minor but important aspect for readers is that color coding is not described. For example, is the red displayed at A91 the carbonyl oxygen?

Figure 3: The structure of V39A reveals a second dimerization interface, presumably induced by the V39A substitution. The new dimer interface is referred to as a weak dimer based on observed amino acid interactions. Quantification in this regard would be helpful (line 230). In addition, it's unclear what's going on at the b/c interface (Fig. 3A). How does this interface compare with crystal packing of the WT protein? Where does the Ni come from and how was it realized that the metal was Ni? How might this interaction relate to calcium? The hydrogen bond in Fig 3D is hard to see and residue 26 in the fig is labeled as glutamine but glycine for line 275.

Why is the CSS score higher for V39A and I91A than WT?

The data indicating the NMR signal to decrease over time (line 318) should be included.

Supple Fig 9 is quite nice and should be included in main Fig 4 with the amino acid labels.

Line 413 requires further explanation; namely that the difference between T2 and T1rho measurements reflect chemical exchange processes.

Fig. 3C & line 481: Is there any evidence in the NMR data for interaction through the ABED face?

Reviewer #2 (Remarks to the Author):

This manuscript is an examination of the involvement of residues at the GFCC' interface of human CEACAM1 in mediating the dimeric and oligomeric interactions, comprising site-directed mutagenesis of 4 residues at the interface, measurements of thermal stability, X-ray crystallography of the mutant proteins and NMR. The significance of the study lies in our incomplete understanding of the monomer-dimer transition of CEACAM1. This is important for understanding how the binding of various proteins to CEACAM1 disrupts the dimer interface to enable heterophilic interactions with proteins like TIM-3, HopQ or Neisseria Opa's. The main findings are that mutation of residues at the interface promotes monomers and highlight a key role for the interface residue, N97. A major strength is the NMR data, showing flexibility of the GFCC' interface. This was performed to a very high standard and is consistent with the biophysical and crystallographic data.

Overall, this is a useful contribution to our understanding of CEACAM1, but the manuscript suffers from a number of problems. Primary among these are over-interpretation of the crystal structures, with seeming avoidance of some critical issues, and also its excess and verbose presentation.

Major points:

This manuscript is excessively long and overly descriptive. The Results section is 13.5 pages ! Much of this could be eliminated to make important details the focus. For example, too much attention is given to each mutant when set against the useful information obtained. Judicious thinning is needed throughout to avoid redundancy and repetition, and eliminate superfluous detail.

Compounding the presentation problem, the pictures in the main manuscript do not show the details being described (hydrogen bonding, hydrophobic packing, structural differences etc). The pictures in the supplementary information do, however, show these details and these should be the main figures.

Specifically, Figs. 2 and 2 do a poor job interpreting the mutants because (a) at the level of zoom given, it's hard to see the residue in question and (b) they don't show a superimposition with the wild-type structure. The superimpositions shown in the Supplemental file should replace the figures in the main text, along with panels at the appropriate level of zoom to show the distances being mentioned.

On the technical side of the manuscript, there are also concerns.

SEC-MALS: There some statements that do not appear congruent with the data in Fig. 1B. Firstly, the curve for the I91A (?) mutant appear bimodal rather than representing a single peak (with the lines being so thin, it's very hard to know which line corresponds to each mutant). And rather than the molecular masses being "varying intermediate" in size between monomer and dimer, it appears more that the mutants are all monomeric with the exception of V39A. Given these points, the summary lines (171-174) may need amending. Also, the apparently intermediate nature of the V39A mutant is interesting, but this is not mentioned. Could this be due to the unique molecular interactions observed for this mutant in the crystal structure?

It is unfortunate that the mutants crystallized in three different space groups because this means that the structural differences observed could arise from crystal packing interactions. Surprisingly, this very important point is seemingly ignored. The same applies to the putative Ni²⁺ seen in the V39A structure because this could also affect structure.

At 3.1Å, as opposed to <2Å, the low resolution of the I91A mutant is notable and yet strangely ignored. Of course, resolution of crystals can vary, but is there a potential structural/crystallographic reason for the lower resolution? This at least merits a mention.

Related to this, the authors should consider whether the relatively high RMSDs for the I91A mutant result from greater uncertainty in this structure due to its lower resolution compared to wild-type. Furthermore, using this structure to infer hydrogen bonding distances and hydrophobic interactions (for Phe29) is circumspect at this resolution (lines 192-193).

Minor points:

A diagram of the domain structure of CEACAM1 would be helpful for the Introduction

Line 160: A correlation of decreasing T_m with decreasing homodimerization affinity is mentioned but where are these data? Is this referring to published data?

It's hard to know what the panel D's of Supp. Fig. 2-4 are showing. There is no comparison with wild-type.

What is the resolution of the published wild-type structure that is presumably being used for the superimpositions (on page 9)? This does not appear to be given.

It sometimes unclear what superimpositions the RMSD's are referring to. Lines 219-222 are good example. An RMSD of 0.677 Å comparing what? Is it the two molecules within one of the dimers (as supposed) or with wild-type?

Stating RMSD values to three decimal places is not justified (e.g. line 187).

Without seeing distances, it's hard to know what is meant by "weaker" hydrogen bonds and hydrophobic interactions.

Lines 204-206 –check wording for repetition.

Line 206 - Suggest to start E99 as a new paragraph.

Suppl. Fig. 8C does not show an ensemble of ten NMR structures but rather a superimposition of one (which one?) with the wild-type crystal structure. Lines 335-336 therefore require amending.

If NMR assignments are 100%, are there no prolines?

Lines 392-393 are not needed.

Line 433 – "except" not "expect"

Lines 487-490 – How might the ABED face participate in oligomeric interactions if it is glycosylated? This only seem possible if there were cellular conditions when the surface is deglycosylated.

Line 619 – "Scattering" not "scatting"

Table 1- Consistency in decimal places is needed. One decimal place for cell dimensions, R factors and Ramachandran numbers is sufficient. The RMSD's for bond lengths and angles for the V39A mutant are too high.

Why were no waters modeled for the sub 2A structures?

(In hindsight, it would have been helpful to see the PDB validation files for these structures)

Dear Reviewers,

We thank the reviewers for their encouraging insights and helpful comments regarding our manuscript entitled “Structural basis of the dynamic human CEACAM1 monomer-dimer equilibrium.” Below we address the reviewers’ comments, concerns and suggestions. The reviewers’ comments below are in bold and our responses are in unbolded text. We believe the reviewers’ comments have enabled us to significantly improve our structural studies and the manuscript.

Reviewers' comments:

Reviewer #1 (Remarks to the Author):

This manuscript merges biophysical and structural biology methods to study single amino acid variants of hCEACAM1 to probe monomer-dimer exchange. The integration of the various techniques is a strength of the manuscript and the NMR studies are particularly compelling. Some aspects require further explanation/consideration, as described in detail below. Overall, the study provides new fundamental information on the impact of interactions at the hCEACAM1 dimerization interface.

We thank the encouraging feedback of Reviewer #1 regarding our NMR studies and integration of various biophysical techniques in the submitted manuscript. We are pleased that reviewer appreciates the important fundamental information our studies provide.

1. Figure 1: Data supporting a single thermal denaturation temperature for each hCEACAM1 IgV variant is not included and probably should be based on text (p. 7, line 157). The claim that decreasing melting temperature correlates with decreasing homodimerization (p. 8, line 160) is similarly not supported. Rather as noted by the authors, the presented data (Fig. 1B) conflicts with the statement; namely, N97A. In addition, the displayed data in Fig. 1B doesn't show a single peak for each protein variant (line 171) – consider for example, I91A which clearly shows two peaks and E99A which can be fitted to two peaks.

Representative raw differential scanning fluorometry spectra for the wildtype and alanine mutants (WT, V39A, E99A, I91A, N97A) depicting the single melting temperature (single inflection point in the melting curve) are now included in the supplemental data as Supplemental Figure 3.

The correlation of melting temperature and contribution to oligomerization affinity for each examined residue (E99, V39, I91 and N97) on CEACAM1 was extrapolated from published studies investigating CEACAM1 homo-oligomerization and GFCC-mediated CEACAM1-CEACAM5 interactions employing ITC and SPR, respectively. The text has been updated to include the appropriate references. The N97A mutant is specifically highlighted in the text as being unique as it has monomerization properties but maintains thermal stability despite not being in dimeric form. This is described in the text as (lines 159-162) , “One exception was the N97A variant that has been reported to be monomeric¹⁶ but exhibited a similar melting temperature (54.09 °C) compared to WT

protein (55.09 °C), suggesting a unique stabilizing property of an alanine at that position and/or promotion of a monomeric state.”

Supplementary Fig. 3. Differential scanning fluorimetry of wildtype (WT) and mutant hCEACAM1 IgV. Relative absorbance emission with corresponding temperature is plotted for WT (black) hCEACAM IgV and V39A (red), I91A (green), N97A (blue), E99A (orange) mutant hCEACAM1 IgV protein samples. Melting temperature (T_M) calculated by first derivative analysis is designated for each curve by colored arrow. Each curve is representative of triplicate samples.

Regarding the presence of multiple peaks in the differential refractive index (dRI) for the I91A and E99A mutant variants, there was only one eluted peak that could be calculated to a specific MW species using the MALS data. We have updated the text to avoid confusion and replaced “peak” with “species” as below (lines 165-171).

“Each hCEACAM1 IgV variant (100 μ M) eluted as a single dominant calculatable molecular weight species but with varying molecular weights ranging from dimer (WT, 23.1 kDa) to monomer (N97A, 13.5 kDa) (Fig. 1B). The presence of a single discernable species for each protein variant and varying intermediate absolute molecular weights suggests rapid rates of exchange between monomeric and dimeric states of the IgV domain rather than a slow equilibrium within the experimental time scale.”

We have also updated the figure to highlight the molecular weight species calculation and the differences in MW calculation for each mutant variant.

Fig. 1. Biophysical characterization of CEACAM1 IgV mutants. Thermal stability and molecular size analysis of hCEACAM1 WT and GFCC' face mutants. (A) Variations in melting point temperature (T_M) determined by differential scanning fluorimetry (DSF) are shown for WT and mutant hCEACAM IgV. (B) Size exclusion chromatography and multi-angle light scattering (SEC-MALS) differential refractive index (dRI) chromatograms and calculated molecular weights are displayed for WT (black), V39A (orange), I91A (green), N97A (blue) and E99A (red).

The reviewer is correct in that there are shoulder/peaks for both the I91A and E99A samples; however, there were no calculable molecular weights that could be determined by the MALS data. One explanation would be that the dRI spectra reflects the elution from the non-analytical, preparative grade separation column that separates proteins and other solutes for MALS measurements. Considering that the MALS measurements could only calculate a specific protein molecular weight for the dominant peak and not the shoulder peak/other peaks, those peaks likely represent either a non-protein species or heterogenous, non-uniform protein species such as a non-specific

2. Figure 2: In comparisons of the crystal structures more clarity is needed. For example does “weaker hydrogen bonded and hydrophobic interactions” (line 192-3) mean lesser? It’s also not possible to see the change in interactions from the figs displayed. Perhaps zoomed in regions are needed along with corresponding electron density maps. The interactions maps (Supp Fig 2B/D for example) would be better as side-by-side comparisons and with corresponding structures/density maps. Referred to interactions/amino acids are in multiple figs not apparent, such as F29 in Fig. 2B (line 202) and hydrogen bonds in Fig 2D (line 207-210). There’s no figure to confidently evaluate the described increase in distance for V39 (line 212). Another minor but important aspect for readers is that color coding is not described. For example, is the red displayed at A91 the carbonyl oxygen?

The reviewer provides many good insights to enhance the comparisons and presentations of the crystal structures to improve clarity. Similar concerns and helpful suggestions were also raised by Reviewer #2. We completely agree with both of the reviewers' recommendations and thus have revised the main text (lines 172-267) of the manuscript with quantifications of the interactions, crystal structures main panel figures and supplemental figures. These changes include:

- We have revised the manuscript with a revised Fig. 3A-D for direct comparison with the WT crystal structure (PDB code 4QXW). As suggested by the reviewers, these comparisons include changes in the hydrophobic and hydrogen bonded interactions of the I91A mutant compared to WT with zoomed in regions and corresponding electron density maps (Fig. 3A-B). These comparisons allowed us to clearly define “weaker hydrogen bonded and hydrophobic interactions” a phrase used earlier in the manuscript by the presence of fewer hydrogen bonds (12 vs 17 for I91A and WT, respectively, Fig. 3D) and weaker hydrophobic interactions as measured by a higher distance between β carbons for F29 and V39 residues in the I91A mutant structure compared to the WT (Fig. 3A-C). Further, we have modified Fig. 3D with some residues in red to highlight the residues involved in the complete loss or decreased number of hydrogen bond

formed in the I91A mutant structure compared to WT. Although we observed differences in the hydrogen bonded and hydrophobic interactions at the GFCC' face between I91A mutant and WT, lower resolution (3.1 Å) of the I91A mutant crystal structure is a possible limitation of these comparisons.

- Similarly, for the E99A mutant crystal structure revised Fig. 2A-D, we provide a similar comparison with the WT crystal structure (PDB code 4QXW) with zoomed in regions and corresponding electron density maps and highlight fewer hydrogen bonds (12 vs 17 for E99A and WT, respectively, Fig 2D) and weaker hydrophobic interactions for F29 and V39 residues. In addition, revised Fig. 2B shows electron densities at the E99A, V39 and G41 residues and depicts loss of critical intermolecular hydrogen bonds observed in the WT structure between the backbone amide of residue G41 with the side chain carboxyl group of residue E99 and also highlights the higher distance of 3.91Å between the β carbons of V39 residues as observed in the E99A mutant structure.
- Revised supplementary Fig. 4C-D provide side-by-side comparison of I91A and E99A residues that mediates hydrogen bonded interactions in the formation of GFCC' face dimer but with fewer hydrogen bonds (12 vs 17 for I99A or E99A and WT).
- As requested by Reviewer #1, the figure legend of each structural figure describes the color coding in the revised manuscript. For example, the color coding scheme of Fig. 2B depicts carbon atoms in magenta, carbonyl oxygen in red and nitrogen in blue, respectively.

We thank the reviewers for these important suggestions. We have updated the text (lines 174-200) in the revised manuscript to read as below .

“To determine the impact of V39, I91, N97, and E99 on hCEACAM1 homodimer formation, we solved the crystal structures of individual V39A, I91A, N97A, and E99A mutant IgV domains to 1.9, 3.1, 1.8, and 1.9 Å resolution, respectively (Table 1), and quantified interactions at GFCC' and ABED faces (Supplementary tables 2-3).

The E99A mutant structure revealed a GFCC' face-mediated homodimer structure (Fig. 2A-D) globally similar to the WT homodimer but with localized conformational differences resulting in a C-alpha root mean square deviation (RMSD) of 0.3 Å (over 1539 atoms) (Fig. 2A, Supplementary Fig. 4A,C). Interestingly, fewer hydrogen bonds (12 vs 17 for E99A and WT, respectively) and weaker hydrophobic interactions were observed for E99A homodimer (Fig. 2A-D, Supplementary Fig. 2A-D, 4A,C). Specifically, side chain to main-chain backbone interactions between E99-G41 were abrogated (Fig. 2A-B, D, Supplementary Fig. 4C) and the intermolecular hydrogen bond network between residues Q89-*Y34*, N97-*Y34*, and Q89-N97 (using nomenclature convention here and after, where Q89 residue is from molecule (a) and *Y34* residue in *italics* is from molecule (b) present in the crystal asymmetric unit) were disrupted at the E99A mutant homodimer interface (Fig. 2D). In addition, the distance between two opposing hydrophobic valine (V39) residues was slightly higher in the E99A homodimer (3.9 Å) compared to WT (3.7 Å) (Fig. 2B-C).

The low resolution (3.1Å) of the I91A mutant structure limits atomic level comparison with the WT homodimer and therefore provides a more global assessment on the structural properties of the I91 residue. The I91A IgV domain adopts a GFCC'-mediated homodimer with RMSD of 0.6 Å (over 1489 atoms) compared to the WT

homodimer (Fig. 3A, Supplementary Fig. 4B,D) with fewer hydrogen bonded (12 vs 17 for I99A and WT, respectively) and weaker hydrophobic interactions (specially for residue F29) (Fig. 3A-D, Supplementary Fig. 2A-D, 4B,D). Thus, the loss of important hydrogen bond interactions and possibly weaker hydrophobic interactions observed in the E99A and I91A mutant structure support the weak dimeric nature of these mutants as observed in our biophysical studies and previous reports^{17,29}.”

Fig. 2. Crystal structure E99A IgV mutant of hCEACAM1. (A) The ribbon diagram of the E99A mutant (magenta) and WT (green) crystal structures with molecules (a) and (b) superimposed on each other. The labeled residues of molecule (a) and molecule (b) are shown by stick representation. The inset shows superimposition of A99 and V39 residues of the E99A mutant (magenta) on E99 and V39 residues of the WT (green), where higher distance of 3.9 Å (magenta) between the 13 carbons of V39 residues was observed in the E99A mutant structure compared to distance of 3.7 Å (green) between the 13 carbons of V39 residues of the WT. (B) The stick representation of E99A, V39 and G41 residues of molecules (a) and (b) with electron densities (2Fo-Fc map at 1.0 σ level) as observed in the E99A mutant structure, which depicts loss of critical intermolecular hydrogen bonds between E99A and G41 as observed in the WT structure between the backbone amide of residue G41 with the side chain carboxyl group of residue E99. The hydrogen bond (3.9 Å) between 13 carbons of V39 residues as observed in the E99A mutant is shown by dashed lines. The carbon atoms in magenta,

carbonyl oxygen in red and nitrogen in blue, are colored, respectively. (C) The arc/stick representation of hydrophobic interactions by F29, I91 and V39 residues of molecules (a) and (b) as observed in the E99A crystal structure (magenta) compared to the WT (green). The residues of molecules (a) and (b) are shown in bold and italics underlined, respectively. The hydrophobic interactions as measured by distance between β carbons of labeled residues are shown in magenta and green for E99A mutant and WT, respectively. The weaker hydrophobic interactions mediated between two V39 residues are shown by fewer pointers on the hydrophobic arc relative to WT. (D) The hydrogen bonded interactions (dashed lines) mediated by GFCC' face labeled residues as observed in the E99A mutant structure. The lesser quantity of hydrogen bonds at GFCC' interface were observed in the E99A mutant structure, 12 (magenta) vs 17 (green) for E99A and WT, respectively. The residues of molecules (a) and (b) are shown in bold and italics underlined, respectively. The residues in red highlight the residues involved in the complete loss or decreased number of hydrogen bond formed in the E99A mutant structure compared to WT. The asterisk (*) indicates formation of two hydrogen bonds (shown by single dashed line) mediated by Q89 residues of molecule (a) and (b) with each other via OE1 and NE2 atoms.

Fig. 3. Crystal structure I91A IgV mutant of hCEACAM1. (A) The ribbon diagram of the I91A mutant (cyan) and WT (green) crystal structures with molecules (a) and (b) superimposed on each other. The inset shows residues of molecule (a) and (b) by stick

representation and superimposition of F29, V39, and A91 residues of the I9AA mutant (cyan) on F29, V39, and I91 residues of the WT (green), where distances between the 13 carbons of the labeled are shown in cyan and green, respectively for I91A mutant and WT. (B) The stick representation of I91A and F29 residues of molecules (a) and (b) with electron densities (2Fo-Fc map at 1.0 σ level) as observed in the I91A mutant structure. The hydrogen bond (7.31 Å) between 13 carbons of F29 residues as observed in the I91A mutant is shown by dashed lines. The carbon atoms in cyan, carbonyl oxygen in red and nitrogen in blue, are colored, respectively. (C) The arc/stick representation of hydrophobic interactions with distance between 13 carbons of labeled residues of molecules (a) and (b) as observed in the I91A crystal structure (cyan) compared to the WT (green). The residues of molecules (a) and (b) are shown in bold and italics underlined, respectively. The weaker hydrophobic interactions mediated between two F29 residues are shown by fewer pointers on the hydrophobic arc relative to WT. (D) The hydrogen bonded interactions (dashed lines) mediated by GFCC' face labeled residues as observed in the I91A mutant structure. The lesser quantity of hydrogen bonds at GFCC' interface were observed in the I91A mutant structure, 12 (cyan) vs 17 (green) for I91A and WT, respectively. The residues of molecules (a) and (b) are shown in bold and italics underlined, respectively. The residues in red indicates loss or decreased number of hydrogen bond interactions in the I91A compared to WT and asterisk (*) indicates two hydrogen bonds mediated by Q89 as described in the Fig.2D.

3. Figure 3: The structure of V39A reveals a second dimerization interface, presumably induced by the V39A substitution. The new dimer interface is referred to as a weak dimer based on observed amino acid interactions. Quantification in this regard would be helpful (line 230). In addition, it's unclear what's going on at the b/c interface (Fig. 3A). How does this interface compare with crystal packing of the WT protein? Where does the Ni come from and how was it realized that the metal was Ni? How might this interaction relate to calcium? The hydrogen bond in Fig 3D is hard to see and residue 26 in the fig is labeled as glutamine but glycine for line 275.

We thank the reviewer for these interesting questions, criticisms and suggestions.

- The reviewer's suggestion regarding quantification of the V39A weak dimer is very helpful. The supplementary tables 2-3 of the revised manuscript address quantifications and strength of the interactions for not only the weak V39A mutant dimer interface, but for all other mutants including I91A, E99A, N97A and compared to the WT. These quantification metrics include number of hydrogen bonds formed at the interface, buried interface area (Å²) and complex significance score (CSS) and collectively form the basis of our comparison among WT and mutant crystal structures as shown in supplementary tables 2-3. For the V39A mutant weak dimer interface, this quantification strategy clearly shows significantly lesser number of hydrogen bonds, and smaller interface area compared to the WT. In addition, the CSS score with the value of 0 for this V39A mutant weak dimer interface compared to the value of 1 of WT highlights weaker GFCC' face interactions and support the intermediate features of the V39A mutant as observed in Fig.1B. To provide more clarity in the quantification scheme of V39A crystal structure, we have updated the V39A crystal structure

main panel Fig. 4A-B with zoomed in V39A site of mutation with electron density and show increased distance between the CC' loops for the V39A weak dimer as measured by a distance of 10.0 Å between V39A 13 carbons compared to distance of 3.7 Å between V39 WT 13 carbons. Further, the revised main panel Fig. 4C shows weaker hydrophobic interactions were observed in the V39A weak dimer interface as measured by higher distance between 13 carbons for F29 (6.7 vs 5.3 Å for V39A vs WT, respectively) and I91 (6.2 / 5.5 Å for V39A vs WT) residues. We hope the revised quantification and figures addresses the reviewer's suggestions.

- We also thank the reviewer for the very interesting question about the interface formed by molecules (b) and (c) in the V39A mutant structure. Our recent analysis (Supplementary Fig. 7) of this interface revealed comparable similarity to a CEACAM1 WT structure with an ABED face dimer (PDB code 2GK2) and C-alpha root mean square deviation (RMSD) of 2.7 Å (over 1647 atoms). In addition similar hydrogen bonded interactions through ABED face residues including N70-Y68, and S72-Y68 were observed in the V39A mutant dimer formed by molecules (b) and (c) and CEACAM1 WT structure (PDB code 2GK2). Taken together, the dimer formed by molecules (b) and (c) showed comparable similarity with previously reported CEACAM1 WT structure with ABED face interactions (PDB code 2GK2).
- Ni⁺⁺ (Nickel) was part of the V39A mutant protein crystallization condition from Index HT screen (Hampton Research), which has 0.005 M Cobalt(II) chloride hexahydrate, 0.005 M Nickel(II) chloride hexahydrate, 0.005 M Cadmium chloride hydrate, and 0.005 M Magnesium chloride hexahydrate. We were very careful in establishing the presence of the metal as Ni⁺⁺ by looking at the electron density, residues involved and monitoring Rfactor/Rfree values during the model building. First, we observed the presence of significant electron density (Supplementary Fig. 6) at the metal site even when 2Fo-Fc map was contoured at 4 sigma (σ) in the molecule (a) and (c). Next, maintaining the occupancy of metal at 0.33 during the refinement, we resolved the final crystal structure with the strategy that showed optimal R/Rfree for bound Ni⁺⁺ (Nickel) compared to Cobalt, Cadmium or Magnesium. Finally, the PDB validation which calculates real space correlation coefficient (RSCC) indicated a very good fit with value of 0.99 for fitted Ni⁺⁺ atoms in the molecule (a) and (c), respectively. Although it was not recognized at the time of the earlier submission, the CEACAM1 WT (PDB code 2GK2) structure was also resolved with bound Ni⁺⁺, which occupies the same atomic position and participates in similar hexadentate interactions with three His105 sidechains and three carbonyl groups of Val106 residues from molecule (a) and its two symmetry mates. Thus, we concluded that Ni⁺⁺ was a better fit (Supplementary Fig. 6). Since calcium was not part of crystallization condition, we didn't try to fit calcium at the observed electron density. Overall, the presence of a Ni⁺⁺ binding site as observed in the V39A and CEACAM1 WT (PDB code 2GK2) structures is very interesting and more studies are needed to determine any functional consequence to human CEACAM1 upon Ni⁺⁺ binding.
- We apologize to the reviewer for the typographical error in previous Fig. 3D of the submitted manuscript. The revised Fig. 5C corrected this error and shows

electron density and zoomed in hydrogen bonded interactions between Q26 of molecules (a) and I67 of molecules (b) as observed in the N97A crystal structure. Further, zoomed in region at the site of N97A mutation with electron density maps is also included in the revised Fig. 5C.

For the reviewer's ease, these above mentioned revised figures, supplemental table 2C, 2H, and 3 with quantifications metrics and residues interactions are included here.

Fig. 4. Crystal structures of the V39A IgV mutant of hCEACAM1. (A) Ribbon diagram (yellow) of the molecules (a), (b), (c), and (d) as observed in the unit cell of the V39A crystal structure. The molecules (a) and (b) make a dimer that mimics a GFCC' face dimer as observed in the hCEACAM1 (PDB code 4QXW) crystal structure (1 inset). The molecules (c) and (d) make a weak GFCC' face dimer where the FG and specifically the CC' loops are far apart and very few GFCC' face residues mediate the interactions (2 inset). The insets show V39A residues of these four molecules by stick representation with electron densities (2Fo-Fc map at 1.0 σ level) and the carbon atoms in yellow, carbonyl oxygen in red and nitrogen in blue, are colored, respectively. In the 1 inset, the distance between 13 carbons of V39A residues of the molecules (a), and (b) is shown by dashed line (4.2 Å), whereas the 2 inset shows increased distance between CC' loops of weak dimer with distance of 10.0 Å between 13 carbons of V39A residues of the molecules (c) and (d). (B) The superimposition of the V39A mutant weak dimer (molecules c and d, colored yellow) and WT (molecules a and b colored green),

whereas CC' loops are further apart in the V39A mutant weak dimer compared to WT as measured by distance between 13 carbons of V39A residues of 10.0 Å in the V39A mutant weak dimer vs 3.7 Å for WT. The inset shows stick representation of the residues F29, A39 and I91 of the weak V39A dimer (yellow) which make weaker hydrophobic interactions compared to WT F29, V39 and I91 residues (green). The A39 and V39 residues of the weak V39A dimer and WT dimer are shown by yellow stick and green stick/solid arrows, representation, respectively. (C) The arc/stick representation of weaker hydrophobic interactions by F29, and I91 residues as observed between molecules (c) and (d) in the formation of weak V39A dimer of the V39A crystal structure relative to WT. The residues of molecules (c) and (d) are shown in bold and italics underlined, respectively. The distances of hydrophobic interactions as measured by distance between 13 carbons of labeled residues are shown in yellow and green for the V39A weak dimer and WT, respectively, and weaker hydrophobic interactions for V39A weak dimer are depicted by fewer pointers on the hydrophobic arc relative to WT.

Fig. 5. V39A weak dimer hydrogen bonded interactions and crystal structure of monomeric N97A IgV mutant. (A) The molecules (c) and (d) that mediate formation of a weak GFCC' face V39A dimer are shown in ribbon and surface representation in yellow, respectively. The residues Y34, Q44, Q89, D94 and N97, which mediate the hydrogen bonded interactions, are shown in stick representation for molecule (c) and surface bright and light red representations for molecule (d). The carbon atoms in

yellow, carbonyl oxygen in red and nitrogen in blue, are colored, respectively. (B) The fewer hydrogen bonded interactions that result in the weak V39A dimer formation between molecules (c) and (d) residues of the V39A mutant crystal structure are shown by dashed lines. The residues of molecules (c) and (d) are shown in bold and italics underlined, respectively. The lesser quantity of hydrogen bonds at GFCC' interface were observed in the V39A weak dimer, 5 (yellow) vs 17 (green) for V39A dimer and WT, respectively. The residues in red indicate loss or decreased number of hydrogen bond interactions in the V39A weak dimer compared to WT. The asterisk (*) indicates formation of two hydrogen bonds (shown by single dashed line) mediated between N97 residue of molecule (c) and Q89 residue of molecule (d). (C) The crystal structure of N97A mutant with two monomeric molecules (a, bottom) and (b, top) shown by ribbon diagram colored silver white. The carbon atoms in silver white, carbonyl oxygen in red and nitrogen in blue, are colored, respectively. The insets 1 and 2 shows stick representation of N97A residues of both molecules with electron density (2Fo-Fc map at 1.0 σ level) and this mutation leads to abrogation of GFCC' face dimer in the N97A crystal structure. The CC' and FG loops are labeled and very limited interface contact between two N97A molecules through ABED face is shown in inset 3, whereas Q26 of molecule (a) and I67 of molecule (b) participates in hydrogen bonded interaction. The two hydrogen bonds of 2.9 Å and 2.9 Å between the aforementioned residues are shown by dashed lines. The residues are shown by stick representation with electron density (2Fo-Fc map at 1.0 σ level).

**Suppl
ment
ary
Fig. 7.**
FG The overall similarity of hCEACAM1 WT structure with an ABED face dimer (PDB code

2GK2) and interface formed by molecules (b) and (c) in the V39A mutant structure. The ribbon diagram of the structural superimposition of the hCEACAM1 WT (PDB code 2GK2) structure (molecules a and b, colored red) and V39A mutant structure (molecules b and c, colored yellow) with with C-alpha root mean square deviation (RMSD) of 2.7 Å (over 1647 atoms). The superimposition revealed an

overall similar minor ABED face contacts through ABED face residues including Y68, N70, and S72 (depicted by arrow). In addition, a similar mode of Ni⁺⁺ binding was observed involving residues His105 in hCEACAM1 WT (PDB code 2GK2) structure (molecules *b*) and V39A mutant structure (molecules *c*). The bound Ni⁺⁺ is shown by sphere and colored red and yellow for hCEACAM1 WT (PDB code 2GK2) and V39A mutant, respectively. The CC' and FG loops are labeled.

Supplementary Table 2: Quantification metrics of human CEACAM1 WT dimer (PDB code 4QXW) and V39A (PDB code 6XNW), I91A (PDB code 6XNT), N97A (PDB code 6XO1), E99A (PDB code 6XNO), and WT (PDB code 2GK2) crystal structures.

Crystal structure	Number of hydrogen bonds	Interface Area (Å ²)	CSS Score
CEACAM1 WT GFCC' face dimer (molecules a/b , PDB 4QXW)	17	824.6	0.9
V39A (molecules a/b)	16	817.5	1.0
V39A (molecules b/c)	4	479.3	0.1
V39A (molecules c/d)	5	525.3	0.0
I91 (molecules a/b)	12	826.0	1.0
N97A (molecules a/b)	N/A	N/A	0.0
E99A (molecules a/b)	12	748.6	0.63
CEACAM1 WT (molecules a/b , PDB code 2GK2)	3	475.4	0.31

Supplementary Table 3C: V39A crystal structure, molecules *b/c*

Interactions	Molecule (b) residues (Interacting atom)	Molecule (c) residues (Interacting atom)	Distance (Å)
1	E5[OE2]	E16[N]	3.5
2	S72[OG]	Y68[OH]	2.9
3	N70[OD1]	Y68[OH]	3.7
4	N70[ND2]	Y68[OH]	3.8

Supplementary Table 3H: Human CEACAM1 (PDB 2GK2), molecules *a/b*

Interactions	Molecule (b) residues (Interacting atom)	Molecule (a) residues (Interacting atom)	Distance (Å)
1	E16[OE1]	N70[ND2]	2.2
2	S72[OG]	Y68[OH]	2.6
3	N70[ND2]	Y68[OH]	2.6

4. Why is the CSS score higher for V39A and I91A than WT?

The reviewer asks a very good question. We calculated the CSS score as part of PDB-PISA analysis which indicates how significant for assembly formation the interface is and any value approaching unity shows a significant interface. So both values of 0.9 or 1 signify an important role of the interface in the dimer formation. We did not find any apparent structural features to explain this small discrepancy. In addition, although WT CSS score was marginally lower than V39A and I91A dimers formed by molecules (a) and (b), respectively, more number of hydrogen bonded interactions and stronger hydrophobic interactions were observed for WT.

5. The data indicating the NMR signal to decrease over time (line 318) should be included.

The NMR spectra that showed wild type CEACAM1 signal decreasing over time unfortunately is complicated by emerging peaks from unfolded proteins over the same time period. We have removed this statement, and provide a figure in this letter for the reviewer's benefit showing NMR signal degradation for N97A CEACAM1 at a more rapid rate (after 2 days). The structural implications of these observations need to be further studied.

6. Supple Fig 9 is quite nice and should be included in main Fig 4 with the amino acid labels.

We thank the reviewer for this suggestion and appreciate reviewer's kind comment. We have included this figure as part of main Figure Fig. 8.

Fig. 8A. Mapping of NMR chemical shift changes of N97A mutant rendered on WT structure. (A) Front and (B) back view of the crystal structure of the hCEACAM1 WT IgV dimer (PDB code 4QXW) with one unit shown in molecular surface representation (white) and another in alpha carbon traces (cyan). The residues with combined NMR chemical shift changes larger than 0.2ppm are colored in red, while residues unassigned in N97A mutant hCEACAM1 IgV are colored in grey.

7. Line 413 requires further explanation; namely that the difference between T2 and T1rho measurements reflect chemical exchange processes.

We have further explained by stating (lines 335-338): “the differences between the transverse relaxation rates R_2 (CPMG) and $R_{1\rho}$ derived R_2^* (spin-lock) showed the same pattern, and clearly confirmed its origin to be from NMR chemical shift exchange processes during relaxation measurements³⁶ (Supplementary Fig. 12B-C).” in the manuscript. Notably, if we substitute R_{20} for R_2^* in approximation, then this difference becomes $R_2(\text{CPMG}) - R_{20} = R_{2ex}$, the residual exchange relaxation rate at the CPMG experimental condition. We also added Supplementary Figure 12C to illustrate this point.

Supplementary Fig.12C. The difference between transverse relaxation rates $R_2(\text{CPMG})$ and $R_2^*(\text{spin-lock})$ (derived from $R_{1\rho}$ according to $R_{1\rho} = R_1 \cos^2 \theta + R_2^* \sin^2 \theta$, where θ is the off-resonance tilt angle of the effective spin-locking field in the rotating frame) of N97A mutant hCEACAM1 IgV. Larger chemical exchanging effect that is insufficiently suppressed by CPMG method mostly occurs for residues at the GFCC' face.

8. Fig. 3C & line 481: Is there any evidence in the NMR data for interaction through the ABED face?

Unfortunately, there is no NMR evidence currently for ABED face interactions. Perhaps it is even more dynamic and weaker.

Reviewer #2 (Remarks to the Author):

This manuscript is an examination of the involvement of residues at the GFCC' interface of human CEACAM1 in mediating the dimeric and oligomeric interactions, comprising site-directed mutagenesis of 4 residues at the interface, measurements of thermal stability, X-ray crystallography of the mutant proteins and NMR. The significance of the study lies in our incomplete understanding of the monomer-dimer transition of CEACAM1. This is important for understanding how the binding of various proteins to CEACAM1 disrupts the dimer interface to enable heterophilic interactions with proteins like TIM-3, HopQ or Neisseria Opa's. The main findings are that mutation of residues at the interface promotes monomers and highlight a key role for the interface residue, N97. A major strength is the NMR data, showing flexibility of the GFCC' interface. This was performed to a very high standard and is consistent with the biophysical and crystallographic data.

Overall, this is a useful contribution to our understanding of CEACAM1, but the manuscript suffers from a number of problems. Primary among these are over-interpretation of the crystal structures, with seeming avoidance of some critical issues, and also its excess and verbose presentation.

We appreciate the encouraging comments of Reviewer #2 regarding the important contribution of our structural study to understand CEACAM1 binding to its various ligands and monomer:dimer dynamic equilibrium. We are glad that Reviewer #2 appreciated the strength of our NMR study and GFCC' face flexibility that we described in the manuscript.

Major points:

1. This manuscript is excessively long and overly descriptive. The Results section is 13.5 pages ! Much of this could be eliminated to make important details the focus. For example, too much attention is given to each mutant when set against the useful information obtained. Judicious thinning is needed throughout to avoid redundancy and repetition, and eliminate superfluous detail.

We thank the reviewer for these criticisms as they allow us to focus on the important details. We have revised the manuscript to avoid redundancy and superfluous detail particularly of the crystal structure section (lines 172-267). We apologize for this and have simplified the text in several areas to eliminate any confusion or ambiguity. We have significantly decreased the size of the manuscript and hope that we have done this to the reviewer's satisfaction.

9. Compounding the presentation problem, the pictures in the main manuscript do not show the details being described (hydrogen bonding, hydrophobic packing, structural differences etc). The pictures in the supplementary information do, however, show these details and these should be the main figures.

We agree with the reviewer. As suggested by the reviewer, we have revised main panel figures with supplementary figures. As shown below, these revised main panel figures provide more clarity and details of the described interactions with zoomed in regions with electron density regions. We hope these changes satisfy the reviewer's concern. We thank the reviewer for these very helpful insights.

3. Specifically, Figs. 2 and 2 do a poor job interpreting the mutants because (a) at the level of zoom given, it's hard to see the residue in question and (b) they don't show a superimposition with the wild-type structure. The superimpositions shown in the Supplemental file should replace the figures in the main text, along with panels at the appropriate level of zoom to show the distances being mentioned.

We thank the reviewer for these important suggestions. We have implemented these suggestions in the revised manuscript and updated main panel figures (Fig. 2A-D, 3A-D, 4A-C, 5A-C) as shown below to incorporate these suggestions.

Fig. 2. Crystal structure E99A IgV mutant of hCEACAM1. (A) The ribbon diagram of the E99A mutant (magenta) and WT (green) crystal structures with molecules (a) and (b) superimposed on each other. The labeled residues of molecule (a) and molecule (b) are shown by stick representation. The inset shows superimposition of A99 and V39 residues of the E99A mutant (magenta) on E99 and V39 residues of the WT (green), where higher distance of 3.9 Å (magenta) between the 13 carbons of V39 residues was observed in the E99A mutant structure compared to distance of 3.7 Å (green) between the 13 carbons of V39 residues of the WT. (B) The stick representation of E99A, V39 and G41 residues of molecules (a) and (b) with electron densities (2Fo-Fc map at 1.0 σ

level) as observed in the E99A mutant structure, which depicts loss of critical intermolecular hydrogen bonds between E99A and G41 as observed in the WT structure between the backbone amide of residue G41 with the side chain carboxyl group of residue E99. The hydrogen bond (3.9 Å) between 13 carbons of V39 residues as observed in the E99A mutant is shown by dashed lines. The carbon atoms in magenta, carbonyl oxygen in red and nitrogen in blue, are colored, respectively. (C) The arc/stick representation of hydrophobic interactions by F29, I91 and V39 residues of molecules (a) and (b) as observed in the E99A crystal structure (magenta) compared to the WT (green). The residues of molecules (a) and (b) are shown in bold and italics underlined, respectively. The hydrophobic interactions as measured by distance between 13 carbons of labeled residues are shown in magenta and green for E99A mutant and WT, respectively. The weaker hydrophobic interactions mediated between two V39 residues are shown by fewer pointers on the hydrophobic arc relative to WT. (D) The hydrogen bonded interactions (dashed lines) mediated by GFCC' face labeled residues as observed in the E99A mutant structure. The lesser quantity of hydrogen bonds at GFCC' interface was observed in the E99A mutant structure, 12 (magenta) vs 17 (green) for E99A and WT, respectively. The residues of molecules (a) and (b) are shown in bold and italics underlined, respectively. The residues in red highlight the residues involved in the complete loss or decreased number of hydrogen bond formed in the E99A mutant structure compared to WT. The asterisk (*) indicates formation of two hydrogen bonds (shown by single dashed line) mediated by Q89 residues of molecule (a) and (b) with each other via OE1 and NE2 atoms.

Fig. 3. Crystal structure I91A IgV mutant of hCEACAM1. (A) The ribbon diagram of the I91A mutant (cyan) and WT (green) crystal structures with molecules (a) and (b) superimposed on each other. The inset shows residues of molecule (a) and (b) by stick representation and superimposition of F29, V39, and A91 residues of the I9AA mutant (cyan) on F29, V39, and I91 residues of the WT (green), where distances between the 13 carbons of the labeled are shown in cyan and green, respectively for I91A mutant and WT. (B) The stick representation of I91A and F29 residues of molecules (a) and (b) with electron densities ($2F_o - F_c$ map at 1.0σ level) as observed in the I91A mutant structure. The hydrogen bond (7.3 Å) between 13 carbons of F29 residues as observed in the I91A mutant is shown by dashed lines. The carbon atoms in cyan, carbonyl oxygen in red and nitrogen in blue, are colored, respectively. (C) The arc/stick representation of hydrophobic interactions with distance between 13 carbons of labeled residues of molecules (a) and (b) as observed in the I91A crystal structure (cyan) compared to the WT (green). The residues of molecules (a) and (b) are shown in bold and italics underlined, respectively. The weaker hydrophobic interactions mediated between two F29 residues are shown by fewer pointers on the hydrophobic arc relative to WT. (D) The hydrogen bonded interactions (dashed lines) mediated by GFCC' face labeled residues as observed in the I91A mutant structure. The lesser quantity of hydrogen bonds at GFCC' interface were observed in the I91A mutant structure, 12 (cyan) vs 17 (green) for I91A and WT, respectively. The residues of molecules (a) and (b) are shown in bold and italics underlined, respectively. The residues in red indicates loss or

decreased number of hydrogen bond interactions in the I91A compared to WT and asterisk (*) indicates two hydrogen bonds mediated by Q89 as described in the Fig.2D.

Fig. 4. Crystal structures of the V39A IgV mutant of hCEACAM1. (A) Ribbon diagram (yellow) of the molecules (a), (b), (c), and (d) as observed in the unit cell of the V39A crystal structure. The molecules (a) and (b) make a dimer that mimics a GFCC' face dimer as observed in the hCEACAM1 (PDB code 4QXW) crystal structure (1 inset). The molecules (c) and (d) make a weak GFCC' face dimer where the FG and specifically the CC' loops are far apart and very few GFCC' face residues mediate the interactions (2 inset). The insets show V39A residues of these four molecules by stick representation with electron densities (2Fo-Fc map at 1.0 σ level) and the carbon atoms in yellow, carbonyl oxygen in red and nitrogen in blue, are colored, respectively. In the 1 inset, the distance between 13 carbons of V39A residues of the molecules (a), and (b) is shown by dashed line (4.2 Å), whereas the 2 inset shows increased distance between CC' loops of weak dimer with distance of 10.0 Å between 13 carbons of V39A residues of the molecules (c) and (d). (B) The superimposition of the V39A mutant weak dimer (molecules c and d, colored yellow) and WT (molecules a and b colored green), whereas CC' loops are further apart in the V39A mutant weak dimer compared to WT as measured by distance between 13 carbons of V39A residues of 10.0 Å in the V39A mutant weak dimer vs 3.7 Å for WT. The inset shows stick representation of the

residues F29, A39 and I91 of the weak V39A dimer (yellow) which make weaker hydrophobic interactions compared to WT F29, V39 and I91 residues (green). The A39 and V39 residues of the weak V39A dimer and WT dimer are shown by yellow stick and green stick/solid arrows, representation, respectively. (C) The arc/stick representation of weaker hydrophobic interactions by F29, and I91 residues as observed between molecules (c) and (d) in the formation of weak V39A dimer of the V39A crystal structure relative to WT. The residues of molecules (c) and (d) are shown in bold and italics underlined, respectively. The distances of hydrophobic interactions as measured by distance between β carbons of labeled residues are shown in yellow and green for the V39A weak dimer and WT, respectively, and weaker hydrophobic interactions for V39A weak dimer are depicted by fewer pointers on the hydrophobic arc relative to WT.

Fig. 5. V39A weak dimer hydrogen bonded interactions and crystal structure of monomeric N97A IgV mutant. (A) The molecules (c) and (d) that mediate formation of a weak GFCC' face V39A dimer are shown in ribbon and surface representation in yellow, respectively. The residues Y34, Q44, Q89, D94 and N97, which mediate the hydrogen bonded interactions, are shown in stick representation for molecule (c) and surface bright and light red representations for molecule (d). The carbon atoms in

yellow, carbonyl oxygen in red and nitrogen in blue, are colored, respectively. (B) The fewer hydrogen bonded interactions that result in the weak V39A dimer formation between molecules (c) and (d) residues of the V39A mutant crystal structure are shown by dashed lines. The residues of molecules (c) and (d) are shown in bold and italics underlined, respectively. The lesser quantity of hydrogen bonds at GFCC' interface were observed in the V39A weak dimer, 5 (yellow) vs 17 (green) for V39A dimer and WT, respectively. The residues in red indicate loss or decreased number of hydrogen bond interactions in the V39A weak dimer compared to WT. The asterisk (*) indicates formation of two hydrogen bonds (shown by single dashed line) mediated between N97 residue of molecule (c) and Q89 residue of molecule (d). (C) The crystal structure of N97A mutant with two monomeric molecules (a, bottom) and (b, top) shown by ribbon diagram colored silver white. The carbon atoms in silver white, carbonyl oxygen in red and nitrogen in blue, are colored, respectively. The insets 1 and 2 shows stick representation of N97A residues of both molecules with electron density (2Fo-Fc map at 1.0 σ level) and this mutation leads to abrogation of GFCC' face dimer in the N97A crystal structure. The CC' and FG loops are labeled and very limited interface contact between two N97A molecules through ABED face is shown in inset 3, whereas Q26 of molecule (a) and I67 of molecule (b) participates in hydrogen bonded interaction. The two hydrogen bonds of 2.9 Å and 2.9 Å between the aforementioned residues are shown by dashed lines. The residues are shown by stick representation with electron density (2Fo-Fc map at 1.0 σ level).

On the technical side of the manuscript, there are also concerns.

4. SEC-MALS: There some statements that do not appear congruent with the data in Fig. 1B. Firstly, the curve for the I91A (?) mutant appear bimodal rather than representing a single peak (with the lines being so thin, it's very hard to know which line corresponds to each mutant). And rather than the molecular masses being "varying intermediate" in size between monomer and dimer, it appears more that the mutants are all monomeric with the exception of V39A. Given these points, the summary lines (171-174) may need amending. Also, the apparently intermediate nature of the V39A mutant is interesting, but this is not mentioned. Could this be due to the unique molecular interactions observed for this mutant in the crystal structure?

The spectra represent refractive index changes (dRI) that are influenced by solute and not necessarily protein. The peak seen with the I9A and E99A samples could therefore represent non-protein species as there was no uniform molecular weight that could be calculated from the MALS measurements. Alternatively, the shoulder peaks could represent various states of aggregation that do not conform to a specific calculatable species. The subtle differences in the intermediate sizes of the I91A, E99A mutant from the monomeric features of the N97A are emphasized in the modified Figure 1A through demarcation of the monomer features/MW of the N97A protein sample as included in the comments to Reviewer #1 who raised similar criticisms. The more pronounced intermediate features of the V39A mutant are indeed intriguing and likely reflect the subtle but important perturbations in the intermolecular interactions at the GFCC face of CEACAM1.

The text (lines 154-171) has been updated to to read “We observed a single T_M for each protein at 25 μ M, suggestive of a single step denaturation event (Supplemental Fig. 3) despite whether the protein was expected to be a hCEACAM1 IgV monomer (N97A) or dimer (WT). There was also a direct correlation of T_M with dimerization affinity of the different hCEACAM1 mutants^{16,17} suggesting that hCEACAM1 IgV homodimerization stabilizes the IgV domain. One exception was the N97A variant that has been reported to be monomeric¹⁶ but exhibited a similar melting temperature (54.09 °C) compared to WT protein (55.09 °C), suggesting a unique stabilizing property of an alanine at that position and/or promotion of a monomeric state. Next, we assayed the solution characteristics of each hCEACAM1 IgV sequence variant by analytical size exclusion chromatography and multi-angle light scattering (SEC-MALS) and calculation of absolute molecular weight. Each hCEACAM1 IgV variant (100 μ M) eluted as a single dominant calculatable molecular weight species but with varying molecular weights ranging from dimer (WT, 23.1 kDa) to monomer (N97A, 13.5 kDa) (Fig. 1B). The presence of a single discernable species for each protein variant and varying intermediate absolute molecular weights suggests rapid rates of exchange between monomeric and dimeric states of the IgV domain rather than a slow equilibrium within the experimental time scale”.

Fig. 1. Biophysical characterization of CEACAM1 IgV mutants. Thermal stability and molecular size analysis of hCEACAM1 WT and GFCC' face mutants. (A) Variations in melting point temperature (T_M) determined by differential scanning fluorimetry (DSF) are shown for WT and mutant hCEACAM IgV. (B) Size exclusion chromatography and multi-angle light scattering (SEC-MALS) differential refractive index (dRI) chromatograms and calculated molecular weights are displayed for WT (black), V39A (orange), I91A (green), N97A (blue) and E99A (red).

Supplementary Fig. 3. Differential scanning fluorimetry of wildtype (WT) and mutant hCEACAM1 IgV. Relative absorbance emission with corresponding temperature is plotted for WT (black) hCEACAM1 IgV and V39A (red), I91A (green), N97A (blue), E99A (orange) mutant hCEACAM1 IgV protein samples. Melting temperature (T_M) calculated by first derivative analysis is designated for each curve by colored arrow. Each curve is representative of triplicate samples.

5. It is unfortunate that the mutants crystallized in three different space groups because this means that the structural differences observed could arise from crystal packing interactions. Surprisingly, this very important point is seemingly ignored. The same applies to the putative Ni^{2+} seen in the V39A structure because this could also affect structure.

We understand the reviewer's concern very well. We propose that it is far more likely that these mutant structures pack in different lattices because their structures are different. If their structures were the same, they would almost certainly crystallize in the same lattice. But as the reviewer noted, one of the most important highlights of this paper is analyses of the residues that promotes monomers and N97A and V39A crystal structures showing different space groups. In addition, although I91A and E99A crystal structure was crystallized in the same space group, different unit cell parameters were observed. But keeping this important suggestion of the reviewer in mind, we have simplified I91A and E99A crystal structure comparisons and highlighted more on the N97A and V39A crystal structures. The observed binding of Ni^{2+} in the V39A structure is very interesting and could be confirmed by us in a previously reported CEACAM1 WT (2GK2) structure (Supplementary Fig. 7). Interestingly, this was observed 2GK2 in the absence of a weak dimer through the GFCC' face. Taken together with SEC-MALS data which suggest the intermediate nature of V39A mutant as noted by the reviewer, we think structural changes observed in the V39A structure reflect more on the V39A mutation than observed Ni^{2+} binding. However, at this point we cannot comment further on the role of Ni^{2+} and its association with human CEACAM1 residues H105 and V106 and implications on the CEACAM1 structure functions.

**Suppl
mentary
Fig. 7.**
FG The
overall
similarity
of
hCEA
CAM1
WT
struc
ture
with
an
ABED
face
dimer
(PDB
code
2GK2)

and interface formed by molecules (b) and (c) in the V39A mutant structure. The ribbon diagram of the structural superimposition of the hCEACAM1 WT (PDB code 2GK2) structure (molecules a and b, colored red) and V39A mutant structure (molecules b and c, colored yellow) with C-alpha root mean square deviation (RMSD) of 2.7 Å (over 1647 atoms). The superimposition revealed an overall similar minor ABED face contacts through ABED face residues including Y68, N70, and S72 (depicted by arrow). In addition, a similar mode of Ni⁺⁺ binding was observed involving residues His105 in hCEACAM1 WT (PDB code 2GK2) structure (molecules b) and V39A mutant structure (molecules c). The bound Ni⁺⁺ is shown by sphere and colored red and yellow for hCEACAM1 WT (PDB code 2GK2) and V39A mutant, respectively. The CC' and FG loops are labeled.

6. At 3.1Å, as opposed to <2Å, the low resolution of the I91A mutant is notable and yet strangely ignored. Of course, resolution of crystals can vary, but is there a potential structural/crystallographic reason for the lower resolution? This at least merits a mention.

We thank the reviewer for this comment and agree with the reviewer's concerns regarding the lower resolution (3.1Å) of the I91A mutant structure. This was one of the reasons we did not perform any conformation and thermal motion analysis of I91A mutant structure. The reviewer's insights on any possible structural/crystallographic reason for the observed lower resolution is interesting and support our thinking that resolution is going to be determined by whatever the protein decides to do when it makes a crystal. To get any insights, we checked whether the I91A mutant structure contains any disordered regions but didn't find any evidence of this. We agree with the reviewer and mentioned the lower resolution of the I91A mutant as possible a limitation of our I91A mutant analysis in the revised manuscript (lines 191-193).

7. Related to this, the authors should consider whether the relatively high RMSDs for the I91A mutant result from greater uncertainty in this structure due to its lower resolution compared to wild-type. Furthermore, using this structure to infer hydrogen bonding distances and hydrophobic interactions (for Phe29) is circumspect at this resolution (lines 192-193).

We agree with the reviewer. As mentioned above, we have updated the manuscript with the lower resolution of the I91A mutant structure as a possible limitation of our study (lines 191-193). However, very clear electron density maps, no electron density or side chain outliers for the residues analyzed and zero Ramachandran outliers for the I91A crystal structure as determined by PDB validation provide more certainty to our analyses.

Minor points:

8. A diagram of the domain structure of CEACAM1 would be helpful for the Introduction

We thank the reviewer for this suggestion. We have revised supplemental Fig.1 as shown below and included the domain structure of CEACAM1 with Ig-V domain.

Supplementary Fig. 1. Domain structure of the hCEACAM1. The human CEACAM1 domain structure (green) contains N-terminal IgV domain followed by three IgC2 domains (A1, B, A2) coupled to a transmembrane sequence and a short or long cytoplasmic tail. The insert shows a ribbon diagram of the IgV domain with two anti-parallel 13-sheet sandwich faces formed by front AGFCC'C" and back BED faces, respectively. The IgV domain of human CEACAM1 mediates GFCC' face homodimer formation and interactions with various ligands such as human TIM-3 and HopQ through the GFCC' face wherein residues of CC' and FG loop are involved in the interactions. The two monomers that form the homodimer are indicated a and b (*italic and underlined*), respectively. The 13 strands and loops are labeled with uppercase letters and *underlined italic* for monomers a and b, respectively.

9. Line 160: A correlation of decreasing T_m with decreasing homodimerization affinity is mentioned but where are these data? Is this referring to published data?

As discussed above, the correlation of melting temperature and contribution to oligomerization affinity for each examined residue (E99, V39, I91 and N97) on CEACAM1 was extrapolated from published studies investigating CEACAM1 homodimerization and GFCC'-mediated CEACAM1-CEACAM5 interactions employing ITC and SPR, respectively. The text has been updated to include the appropriate references (lines 157-159).

0. It's hard to know what the panel D's of Supp. Fig. 2-4 are showing. There is no comparison with wild-type.

We apologize to the reviewer for this oversight and lack of clarity. As suggested by the reviewer and shown above, we have moved this part of supplementary figures to the main panel figures and revised the main panel Figs. 2C, 3C, and 4C to provide hydrophobic distance comparisons with WT. In addition, zoomed in regions and electron density maps are provided for better clarity and help in the understanding of the observed changes.

1. What is the resolution of the published wild-type structure that is presumably being used for the superimpositions (on page 9)? This does not appear to be given.

We apologize to the reviewer for this oversight. The resolution of the WT human CEACAM1 structure is 2.04 Å. We have updated the text (lines 87-92) in the revised manuscript to read "As demonstrated in our previously reported high resolution (2.04 Å) crystal structure of the wildtype (WT) CEACAM1 homodimer (PDB code 4QXW),⁸ the side chains of residues S32, Y34, Q44, Q89, N97, and E99 form a hydrogen bonding network at the GFCC' interface that includes additional side-chain to main-chain backbone interactions between S32 to L95, Q44 to N97, and E99 to G41 and hydrophobic interactions by residues F29, V39, and I91 (Supplementary Fig. 2A-D)."

2. It sometimes unclear what superimpositions the RMSD's are referring to. Lines 219-222 are good example. An RMSD of 0.677 Å comparing what? Is it the two molecules within one of the dimers (as supposed) or with wild-type?

We thank the reviewer for this very helpful insight. We have revised the manuscript throughout to provide more clarity with RMSD comparisons.

3. Stating RMSD values to three decimal places is not justified (e.g. line 187).

We agree with the reviewer and revised RMSD values to the first decimal places in the revised manuscript.

4. Without seeing distances, it's hard to know what is meant by "weaker" hydrogen bonds and hydrophobic interactions.

We thank the reviewer for this very important suggestion. To provide clarity in the observed weaker hydrogen bonds and hydrophobic interactions, we have updated the main panel crystal figures to include distances of the observed change and provided

zoomed in figures. We hope these changes along with supplementary table 3 shown below that provide hydrogen bonded distances for all the crystal structures analyzed provide better clarity in the revised manuscript.

Supplementary Table 3 : Hydrogen bonded interactions as observed in human CEACAM1 WT dimer (PDB code 4QXW) and V39A (PDB code 6XNW), I91A (PDB code 6XNT), N97A (PDB code 6XO1), E99A (PDB code 6XNO) and CEACAM1 WT (PDB code 2GK2) crystal structures.

Table 3A: Human CEACAM1 WT dimer, molecules a/b (PDB 4QXW)

Interactions	Molecule (a) residues (Interacting atom)	Molecule (b) residues (Interacting atom)	Distance (Å)
1	L95[O]	S32[OG]	3.0
2	E99[OE2]	G41[N]	2.9
3	L95[O]	Q44[NE2]	2.8
4	Y34[OH]	Q89[NE2]	3.3
5	Q89[OE1]	Q89[NE2]	3.0
6	Q44[OE1]	N97[N]	3.0
7	S32[OG]	N97[ND2]	2.9
8	Y34[OH]	N97[ND2]	3.0
9	Q89[OE1]	N97[ND2]	3.8
10	S32[OG]	L95[O]	2.9
11	G41[N]	E99[OE1]	2.7
12	Q44[NE2]	L95[O]	2.8
13	Q89[NE2]	Q89[OE1]	3.0
14	Q89[NE2]	Y34[OH]	3.6
15	N97[N]	Q44[OE1]	3.2
16	N97[ND2]	S32[OG]	3.0
17	N97[ND2]	Y34[OH]	3.5

Table 3B: V39A mutant crystal structure, molecules a/b

Interactions	Molecule (a) residues (Interacting atom)	Molecule (b) residues (Interacting atom)	Distance (Å)
--------------	---	---	--------------

1	L95[O]	S32[OG]	2.9
2	E37[O]	R38[NH1]	3.1
3	E99[OE1]	G41[N]	2.7
4	L95[O]	Q44[NE2]	3.0
5	Q89[OE1]	Q89[NE2]	3.0
6	Q44[OE1]	N97[N]	3.0
7	S32[OG]	N97[ND2]	3.5
8	S32[OG]	L95[O]	3.1
9	R38[NH1]	E37[O]	3.9
10	G41[N]	E99[OE1]	2.7
11	Q44[NE2]	L95[O]	2.8
12	Q89[NE2]	Y34[OH]	3.3
13	Q89[NE2]	Q89[OE1]	3.1
14	N97[N]	Q44[OE1]	3.0
15	N97[ND2]	S32[OG]	3.3
16	N97[ND2]	Y34[OH]	3.2

Table 3C: V39A mutant crystal structure, molecules b/c

Interactions	Molecule (b) residues (Interacting atom)	Molecule (c) residues (Interacting atom)	Distance (Å)
1	E5[OE2]	E16[N]	3.5
2	S72[OG]	Y68[OH]	2.9
3	N70[OD1]	Y68[OH]	3.7
4	N70[ND2]	Y68[OH]	3.8

Table 3D: V39A mutant crystal structure, molecules c/d

Interactions	Molecule (c) residues (Interacting atom)	Molecule (d) residues (Interacting atom)	Distance (Å)
1	D 94[O]	Q44[NE2]	3.1
2	N97[OD1]	Q89[NE2]	3.3

3	Q44[NE2]	D94[O]	2.9
4	N97[ND2]	Q89[OE1]	2.8
5	N97[ND2]	N97[OD1]	2.4

Table 3E: I91A mutant crystal structure, molecules *a/b*

Interactions	Molecule (a) residues (Interacting atom)	Molecule (b) residues (Interacting atom)	Distance (Å)
1	L95[O]	S32[OG]	3.1
2	E99[OE1]	G41[N]	3.2
3	L95[O]	Q44[NE2]	2.8
4	Q89[OE1]	Q89[NE2]	2.9
5	Q89[OE1]	N97[ND2]	3.4
6	S32[OG]	L95[O]	2.6
7	G41[N]	E99[OE1]	2.9
8	Q44[NE2]	L95[O]	3.0
9	Q89[NE2]	Q89[OE1]	2.8
10	N 97[N]	Q44[OE1]	3.8
11	N97[ND2]	S32[OG]	3.9
12	N97[ND2]	Q89[OE1]	3.6

Table 3F: N97A mutant crystal structure, molecules *a/b*

Interactions	Molecule (a) residues (Interacting atom)	Molecule (b) residues (Interacting atom)	Distance (Å)
1	N26[OE1]	I67[N]	2.9
2	N26[NE2]	I67[O]	2.9
3	N53[NE2]	G58[O]	3.9
4	N53[NE2]	N61[O]	3.5

Table 3G: E99A mutant crystal structure, molecules *a/b*

Interactions	Molecule (a) residues	Molecule (b) residues	Distance (Å)
--------------	-----------------------	-----------------------	--------------

	(Interacting atom)	(Interacting atom)	
1	S32[OG]	N97[ND2]	3.0
2	Y34[OH]	N97[ND2]	3.3
3	Y34[OH]	Q89[NE2]	3.6
4	Q44[OE1]	N97[N]	3.1
5	Q89[OE1]	Q89[NE2]	3.2
6	L95[O]	S32[OG]	2.9
7	L95[O]	Q44[NE2]	2.8
8	S32[OG]	L95[O]	3.0
9	Q 44[NE2]	L95[O]	2.8
10	Q89[NE2]	Q89[OE1]	3.0
11	N97[N]	Q44[OE1]	3.4
12	N97[ND2]	S32[OG]	3.3

Table 3H: Human CEACAM1 (PDB 2GK2), molecules *a/b*

Interactions	Molecule (b) residues (Interacting atom)	Molecule (a) residues (Interacting atom)	Distance (Å)
1	E16[OE1]	N70[ND2]	2.2
2	S72[OG]	Y68[OH]	2.6
3	N70[ND2]	Y68[OH]	2.6

15. Lines 204-206 –check wording for repetition.

We thank the reviewer for this suggestion. We have updated the text (lines 197-200) in the revised manuscript to read “Thus, the loss of important hydrogen bond interactions and possibly weaker hydrophobic interactions observed in the E99A and I91A mutant structure support the weak dimeric nature of these mutants as observed in our biophysical studies and previous reports^{17,29}.”

16. Line 206 - Suggest to start E99 as a new paragraph.

We agree with the reviewer and E99 crystal structure description and comparison starts as a new paragraph.

17. Suppl. Fig. 8C does not show an ensemble of ten NMR structures but rather a superimposition of one (which one?) with the wild-type crystal structure. Lines 335-336 therefore require amending.

We have modified the figure to show the NMR ensemble.

Supplementary Fig. 11. WT NMR secondary structure prediction and comparison. Similarity and superimposition of human CEACAM1 WT predicted NMR structure (cyan, BMRB ID 50368) and crystal structure (green, PDB code 4QXW). (A) The top ten lowest energy NMR predicted structures of hCEACAM1 WT (cyan) superimposition with the WT crystal structure (green) with a RMSD of 0.7 Å (over 664 atoms). All the ten NMR states are shown. The CC' and FG loops are labeled. (B) The superimposition of one of the NMR predicted structure of hCEACAM1 WT (cyan) superimposition with the WT crystal structure (green). Only one state is shown.

18. If NMR assignments are 100%, are there no prolines?

We have modified the manuscript to state that “NMR assignments are 100% (excluding five proline residues)”.

19. Lines 392-393 are not needed.

We agree with the reviewer. We have removed these lines in the revised manuscript.

20. Line 433 – “except” not “expect”

We have corrected this typographical error in the revised manuscript.

0. Lines 487-490 – How might the ABED face participate in oligomeric interactions if it is glycosylated? This only seem possible if there were cellular conditions when the surface is deglycosylated.

Reviewer asked a very interesting question. The human CEACAM1 Ig-V domain has three glycosylation sites at the ABED face including residues N70, N77, and N81. The crystal structure of CEACAM1 WT (PDB 2GK2) showed minor interactions through the ABED face involving residues such as Y68, N70, and S72 (Supplemental table 3H) and glycosylation sites could affect this ABED face interactions. We also observed a similar ABED face with minor interactions formed by V39A (b) and (c) molecules in our reported study (Supplemental table 3C). In addition, we also observed minor

interactions through ABED face in N97A mutant crystal structure but mediated mainly by residues I67 and Q26 (Supplemental table 3F). Overall, involvement of different residues in PDB 2GK2 (or V39A (b) and (c) interface) and N97A mutant ABED face suggest flexibility in minor interactions through ABED face. Unfortunately, more biophysical and structural studies are needed to exactly probe the role of glycosylaton on the ABED face mediated oligomeric association.

Table 3H: Human CEACAM1 (PDB 2GK2), molecules *a/b*

Interactions	Molecule (b) residues (Interacting atom)	Molecule (a) residues (Interacting atom)	Distance (Å)
1	E16[OE1]	N70[ND2]	2.2
2	S72[OG]	Y68[OH]	2.6
3	N70[ND2]	Y68[OH]	2.6

Table 3C: V39A mutant crystal structure, molecules *b/c*

Interactions	Molecule (b) residues (Interacting atom)	Molecule (c) residues (Interacting atom)	Distance (Å)
1	E5[OE2]	E16[N]	3.5
2	S72[OG]	Y68[OH]	2.9
3	N70[OD1]	Y68[OH]	3.7
4	N70[ND2]	Y68[OH]	3.8

Table 3F: N97A mutant crystal structure, molecules *a/b*

Interactions	Molecule (a) residues (Interacting atom)	Molecule (b) residues (Interacting atom)	Distance (Å)
1	N26[OE1]	I67[N]	2.9
2	N26[NE2]	I67[O]	2.9
3	N53[NE2]	G58[O]	3.9
4	N53[NE2]	N61[O]	3.5

22. Line 619 – “Scattering” not “scatting”

We have corrected this typographical error in the revised manuscript.

23. Table 1- Consistency in decimal places is needed. One decimal place for cell

dimensions, R factors and Ramachandran numbers is sufficient. The RMSD's for bond lengths and angles for the V39A mutant are too high.

We have incorporated these suggestions in the revised manuscript. In addition, we refined the V39A crystal structure to obtain lower bond length RMSD's for bond length and bond angle which helped in the better PDB validation. We thank reviewer for very helpful suggestion.

Table 1: Crystal information, data collection and refinement parameters

	V39A (PDB code 6XNW)	I91A (PDB code 6XNT)	E99A (PDB code 6XNO)	N97A (PDB code 6XO1)
Data collection statistics				
Space Group	P 3	P 4212	P 4212	C 2221
Cell constants a (Å), b (Å), c (Å), α° , β° , γ°	91.4, 91.4, 64.4, 90.0,	102.1, 102.1, 61.0, 90.0, 90.0, 90.00	106.8, 106.8, 62.2, 90.0, 90.0, 90.00	55.9,56.8,124.5, 90.0, 90.0, 120.0
Resolution (Å)*	39.59-1.90 (1.94-1.90)	72.21-3.1 (3.31-3.1)	75.5-1.9(1.94-1.90)	28.760-1.76 (1.8-1.76)
No. of measurements*	366913 (23237)	73320 (12990)	393727(24522)	231087 (6712)
Unique reflections*	47459(3048)	6265 (1107)	28973 (1843)	19429 (1087)
I/sigma I*	7.8 (1.4)	9.4 (3.8)	14.3 (2.3)	19.5 (2.0)
Completeness (%)*	100 (100)	100 (100)	100 (100)	96.1(73.2)
Redundancy*	7.7 (7.6)	11.7 (11.7)	13.6 (13.3)	11.9 (6.2)
R_{merge} (%) a,-	18.8 (155.1)	24.6(75.4)	11.9 (140.3)	8.6 (833)
$CC_{1/2}$ *	0.992 (0.304)	0.963 (0.894)	0.998 (0.769)	0.998 (0.794)
Structure refinement				
R_{work} (%) ^{b,*}	14.4	22.1	18.9	19.1
R_{free} (%) ^{c,*}	18.6	25.8	22.3	23.9
No. atoms				
Protein/Water	3373	1720	1814	1870
R.m.s.deviations				
Bond-lengths (Å)	0.021	0.014	0.021	0.019
Bond-angles (°)	2.27	1.85	2.15	1.91
Ramachandran				

Favored regions (%)	95.0	94.3	96.7	97.1
Allowed regions (%)	4.1	4.8	2.4	2.9
Disallowed (%)	0.9	0.9	0.9	0
B-factors, All atoms (Å²)	28	42.0	31.0	16.0

^a $R_{merge} = \sum |I - \langle I \rangle| / \sum I$, where I is the observed intensity and $\langle I \rangle$ is the weighted mean of the reflection intensity.

^b $R_{work} = \sum ||F_o| - F_c| / \sum |F_o|$, where F_o and F_c are the observed and calculated structure factor amplitudes, respectively.

^c R_{free} is the crystallographic R_{work} calculated with 5 % of the data that were excluded from the structure refinement.

* Values in the parentheses are for highest resolution shell.

24. Why were no waters modeled for the sub 2A structures?

(In hindsight, it would have been helpful to see the PDB validation files for these structures)

We thank the reviewer. We modeled 4 water molecules in I91A mutant structure. We provide PDB validation reports of all the crystal structure along with the revised manuscript.

REVIEWERS' COMMENTS:

Reviewer #1 (Remarks to the Author):

This manuscript is much improved; the authors elegantly merge multiple methods to make important insights into CEACAM1 activity. All of my concerns have been addressed.